# ON THE LOCAL COMPLEXITY OF LINEAR REGIONS IN DEEP ReLU NETWORKS

## ABSTRACT

We define the *local complexity* of a neural network with continuous piecewise linear activations as a measure of the density of linear regions over an input data distribution. We show theoretically that ReLU networks that learn low-dimensional feature representations have a lower local complexity. This allows us to connect recent empirical observations on feature learning at the level of the weight matrices with concrete properties of the learned functions. In particular, we show that the local complexity serves as an upper bound on the total variation of the function over the input data distribution and thus that feature learning can be related to adversarial robustness. Lastly, we consider how optimization drives ReLU networks towards solutions with lower local complexity. Overall, this work contributes a theoretical framework towards relating geometric properties of ReLU networks to different aspects of learning such as feature learning and representation cost.

## 1 INTRODUCTION

Despite the numerous achievements of deep learning, many of the mechanisms by which deep neural networks learn and generalize remain unclear. An "Occam's Razor" style heuristic is that we want our neural network to parameterize a simple solution after training, but it can be challenging to establish a useful metric of the complexity of a deep neural network (Hu et al., 2021). A growing body of research has sought to gain insights into the complexity of deep neural networks in the case where we use piece-wise linear activation functions, such as ReLU, LeakyReLU, or Maxout. If $\phi$ is a continuous piecewise linear (CPWL) activation function and $A_i(x) = W_i x - \beta_i$ is a parameterized affine linear function, $i = 1, \ldots, L$, we consider a network of the following form:

$$\mathcal{N}_\theta(x) = A_L \circ \phi \circ A_{L-1} \cdots \phi \circ A_1(x), \quad x \in \mathbb{R}^{n_0}. \tag{1}$$

The network function $\mathcal{N}_\theta \colon \mathbb{R}^{n_0} \to \mathbb{R}^{n_L}$ parameterized by $\theta = (W_i, \beta_i)_i$ is then also be a CPWL function. For any fixed choice of the parameter $\theta$, the input domain $\mathbb{R}^{n_0}$ is partitioned into *linear regions* where the function is linear. These partitions of the input space have been used extensively to study diverse topics such as the expressive power, decision boundaries in classification, gradients and parameter initialization, and generalization (e.g., Montúfar et al., 2014; Raghu et al., 2017; Zhang et al., 2018; Balestriero & Baraniuk, 2018; Grigsby & Lindsey, 2022; Brandenburg et al., 2024; Telgarsky, 2016). In this work we aim to advance a theoretical framework towards better understanding the local distribution of linear regions near the data distribution and how it relates to other relevant aspects of learning such as robustness and representation learning.

### 1.1 MOTIVATION

In the kernel regime, neural networks with piecewise linear activations are observed to follow lazy training (Chizat et al., 2019) and bias towards smooth interpolants which do not significantly change the structure of linear regions during training (see, e.g., Williams et al., 2019; Jin & Montúfar, 2023). On the other hand, for networks in the active regime, which are not well approximated by linearized models, one observes significant movement of the linear regions and in some cases a bias towards interpolants that have only a small number of linear regions (e.g., Maennel et al., 2018; Williams et al., 2019; Shevchenko et al., 2022). Characterizing the dynamics of linear regions at a theoretical level remains a significant outstanding challenge, even for shallow networks. Recent empirical studies have demonstrated interesting dynamics of the linear regions near the training data points.

In particular, Humayun et al. (2024b) have shown that in the terminal phase of training, the number of linear regions near the data drops significantly, and this drop corresponds to an increase in the model's adversarial robustness. We replicate similar experiments in Figure 3. Related is the concept of "grokking" which refers to the sudden improvement in the generalization error or robustness after extended training periods, often long after the training loss has reached near-zero values. Grokking has been associated with representation learning, where an emerging idea is that late generalization may occur if and when a network learns the "right" representation for the task at hand (Liu et al., 2022). In particular, some works have claimed that networks learn low-dimensional representations during grokking (Fan et al., 2024; Yunis et al., 2024b;a). This motivates us to consider the following:

***Question 1:*** *How does representation learning relate to the complexity of linear regions?*

To better understand representation learning, we consider the dimension of the feature manifold as measured by the average rank of the Jacobian of the intermediate layer representations with respect to the input. In particular, based on the structure of various theoretical bounds (Montúfar, 2017; Serra et al., 2018; Hinz, 2021), we expect that networks that learn low-dimensional feature manifolds will generally also have fewer linear regions. Empirical results also show that networks which undergo a drop in the number of linear regions tend to be much simpler, having a nearly piecewise constant structure, hinting at a connection between the local distribution of the linear regions and the global structure of the learned function (Humayun et al., 2024b). Related is the concept of "neural collapse", which refers to a phenomenon where, in the terminal phase of training, the within-class variance of the last layer features tends towards zero (Papyan et al., 2020). Furthermore, prior literature has suggested a connection between the size of linear regions and robustness (Croce et al., 2019). Thus, a natural question we concern ourselves with is:

***Question 2:*** *Can we connect the local density of linear regions to the robustness of a network?*

We attempt to answer this question by comparing a measure of the local density of linear regions to the total variation the a network over the input space. Aspects in this direction have appeared in context of parameter initialization and the gradients of a network with respect to its inputs (e.g., Hanin & Rolnick, 2018; Tseran & Montúfar, 2023). Lastly, we are interested in the relation between parameters and functions, and how optimization may cause networks to converge to solutions with lower complexity in terms of linear regions. To this end we compare our measure of local complexity to the distribution of parameters, building on ideas that have been used to study the expected number of linear regions (Hanin & Rolnick, 2019b), and the representation cost of a network, a quantity which has been previously linked to sparsity of weight matrices (Jacot, 2023a).

## 1.2 CONTRIBUTIONS

This work takes steps towards establishing quantitative links in ReLU networks between the distribution of linear regions in the input space, representation learning, and parameter optimization:

- We introduce a framework for understanding model complexity based on the linear regions over the input space. In Section 3 we define the *local complexity* (LC) as the average density of non-linearites over a dataset. To capture the typical behavior of the functions, we define this measure in a way that is robust to perturbations of the bias parameters.

- In Section 4 we establish theoretical connections between the proposed local complexity and the *local rank*, which we define as the average dimension of the feature manifold at intermediate layers. This offers a link between the network complexity and representation learning.

- In Section 5 we demonstrate a bound between the local complexity and the total variation of a network over the input space. This offers a possible path towards understanding how the linear regions can relate to adversarial robustness and phenomena like neural collapse.

- We explore links between local complexity and parameter optimization. In Section 6 we show that the local complexity is bounded by the representation cost and by the ranks of the weight matrices. As a consequence, we can relate the density of linear regions to results on the implicit regularization of the ranks of weight matrices.

## 2 RELATED WORKS

Several works have studied bounds on the number of linear regions of the functions represented by deep ReLU networks (e.g., Pascanu et al., 2014; Montúfar et al., 2014; Serra et al., 2018). For deep neural networks the maximum number of linear regions will typically be polynomial in the width and exponential in the input dimension and number of layers. However, the parameters that achieve this upper bound typically occupy only a small region of the parameter space. In fact, if one considers the expected number of linear regions over a probability distribution of parameters that satisfies certain reasonable conditions, one finds that this is bounded above by the number of neurons raised to the input dimension (Hanin & Rolnick, 2019b;a; Tseran & Montúfar, 2021). In other words, for a random choice of the parameters, one is more likely to see a number of linear regions that is much smaller than the hard upper bound. Some works have analyzed the distortion length of curves in the input space by ReLU networks (Raghu et al., 2017). Hanin et al. (2022) looks at how the expected length of these curves changes from input to output of a network. In a similar vein, Goujon et al. (2024) estimate the typical number of non-linearity points encountered by a 1D curve in the input space. Other works have studied the effect that the architecture may have on the geometry and the topology of decision boundaries in classification (Zhang et al., 2018; Grigsby & Lindsey, 2022; Alfarra et al., 2023; Brandenburg et al., 2024).

A few works have tried to understand the local behavior and the dynamics of linear regions during training. In particular, Humayun et al. (2024b) compare the phenomenon of grokking to a simplification of the linear regions near the training data points. They demonstrate empirically that during the terminal phase of training, a relatively sudden drop in the number of linear regions corresponds to an improvement in the model's adversarial robustness. Cohan et al. (2022) study the evolution of linear regions in the state space of networks trained for deep reinforcement learning, finding a decrease in the density during training, as measured by trajectories in the state space. In a related work, Zhu et al. (2020) derive an algorithm for computing an upper bound on the number of linear regions near a data point and look into the training dynamics of the linear regions. Sattelberg et al. (2023) examine the linear regions local to a dataset of trained networks and note that they tend to be relatively simple. Croce et al. (2019) relate the size of linear regions to adversarial robustness. Another result that links complexity of the linear regions to robustness is that of Humayun et al. (2023b), which leverages the linear region structure of ReLU networks to design an algorithm which improves adversarial robustness. Similar to some of our results, Li et al. (2022) relate adversarial robustness to model complexity as defined by the VC dimension. In a similar flavor to our definition of the local complexity, Gamba et al. (2022) build a complexity measure related to linear regions and propose that the exact number of linear regions may not be the best metric for model complexity, preferring instead to focus on a more robust measure. Other works have analyzed 'knot' points, or non-linearity points through an optimization perspective, where in particular Shevchenko et al. (2022) obtain a bound on the number of knots between training inputs for univariate shallow ReLU networks in the mean-field regime (Mei et al., 2018; Rotskoff & Vanden-Eijnden, 2022).

We may also highlight a few of the works that look at the dimensionality of representations, such as those of Humayun et al. (2024a); Jacot (2023a;b); Jacot et al. (2024); Scarvelis & Solomon (2024). Our definition of local rank bears similarity to that of Humayun et al. (2024a) and Patel & Shwartz-Ziv (2024) as well as the "Jacobian rank" introduced by Jacot (2023a). The low-rank bias of neural networks is a related idea that has been studied by Súkeník et al. (2024) and Timor et al. (2023). Several other works in this area have sought to characterize the dimension of data manifolds through the use of diffusion models (Stanczuk et al., 2022). A connection between the rank of learned embeddings and the representation cost was demonstrated in the work of Jacot (2023a). The papers of Dherin et al. (2022); Munn et al. (2024) highlight connections between neural collapse and a quantity they called the "geometric complexity", which is generally reminiscent of the Dirichlet energy. Our definition of the total variation of a network over the data distribution is bears resemblance to their definition of the geometric complexity.

## 3 THE LOCAL COMPLEXITY OF ReLU NETWORKS

We first aim to define a notion that captures the density of linear regions locally near a given dataset. We will consider ReLU networks defined as in (1), with input dimension $n_0$, hidden layers of widths $n_1, \ldots, n_{L-1}$, and output dimension $n_L = 1$. Given a fixed parameter $\theta$ and an input $x$, the $\ell$th layer

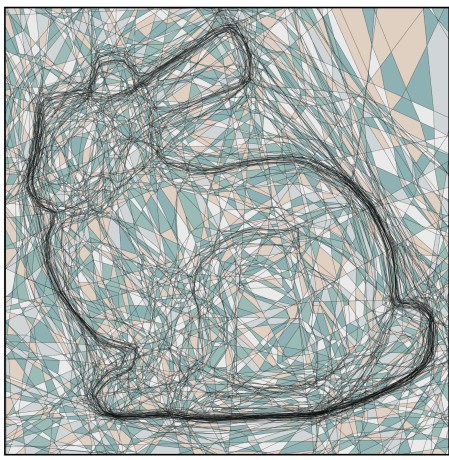 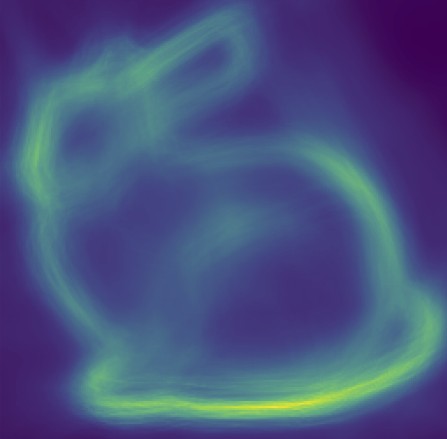

Figure 1: Left: Linear regions of a trained neural network over a two-dimensional input domain. The nonlinear locus is shown in black and linear regions are colored at random. Right: Heat map of the numerically estimated local complexity density function $f(x)$ over the same domain. Precise details are provided in Appendix B.1. The figure illustrates that our definition of local complexity, as well as the equations derived in Theorem 2, are consistent with our intuitive interpretation of this quantity as a local density of non-linearity over the input space.

feature vector pre-activation is given by $v_\ell(x) = A_\ell \circ \phi \circ A_{\ell-1} \cdots \phi \circ A_1(x) \in \mathbb{R}^{n_\ell}$. The array of sign vectors $[\mathrm{sgn}(v_\ell(x))]_{\ell=1}^{L-1}$ is called the *activation pattern* for the input $x$, and the set of all inputs that share the same activation pattern is the corresponding *activation region* in input space, for the given parameter $\theta$. For each fixed parameter value, the function $\mathcal{N}_\theta$ has a constant slope over each activation region. We make the mild assumption that no two activation regions whose activation pattern differ by one neuron will share the same slope. This is a generic property that holds true for all parameters except for a zero Lebesgue measure subset (see, e.g., Hanin & Rolnick, 2019b; Grigsby & Lindsey, 2022) and implies that the activation regions coincide with the linear regions. For fixed parameter $\theta$, the *nonlinear locus* of the network $\mathcal{N}_\theta$ over the input space is given by

$$\mathcal{B}_{\mathcal{N}_\theta} = \{x \in \mathbb{R}^{n_0} : \nabla_x \mathcal{N}_\theta(\cdot) \text{ is discontinuous at } x\}. \tag{2}$$

The nonlinear locus separates the input space into linear regions, over which the function is linear.

### 3.1 THE ROLE OF NOISE IN COMPUTING LOCAL COMPLEXITY

We seek to define a measure for the local density of linear regions that are robust to small perturbations of the weights. We take the view that the average number of linear regions over a local region of parameter space can be more meaningful that the number of regions at a fixed parameter value. Thus, we wish to understand $\mathcal{B}_{\mathcal{N}_\theta}$ as a random object given a choice of weight matrices.

Tracking the number of linear regions for particular parameters requires that one solves a system of parametric equations of the form $z_{\ell,i}(x) = \beta_{\ell,i}$, which can be difficult. On the other hand, examples from algebraic geometry suggest that tracking expected values of the number of solutions to parametric systems can be easier (Malajovich, 2023). This approach and the resulting proof techniques bear resemblance to the application of the co-area formula in the work of Hanin & Rolnick (2019b) or the Kac-Rice formula, which is known for characterizing the size of level sets in random fields (Berzin et al., 2022). In contrast to the definitions of Hanin & Rolnick (2019b), we will consider the distribution of linear regions over the input space and the behavior depending on specific parameters, which are aspects that are not covered in their work. While we do not directly apply the Kac-Rice formula, we find its structure and applications to be conceptually relevant.

Here and in the following we will write the pre-activation of the $i$th unit at the $\ell$th layer as $v_{\ell,i}(x) = z_{\ell,i}(x) - \beta_{\ell,i}$, where $\beta_{\ell,i}$ is the bias. The simplest model that allows us to consider expected values is to introduce additive noise $\delta_{\ell,i}$ and track the 0 level set of $v_{\ell,i}(x) + \delta_{\ell,i}$. It is possible to introduce

additive noise to both biases and weights, but we will focus on the biases since these only translate the activation boundaries, whereas the weights affect both the position and the orientation of the activation boundaries. In Appendix B we provide numerical illustrations showcasing the effects of adding noise either only to the biases or adding noise to both the biases and the weights and how both models produce qualitatively similar results.

Let $\theta = (W_1, \beta_1, W_2, \beta_2, \ldots, W_L, \beta_L)$ be a particular choice of parameters. Consider then the parameters with noisy biases $\tilde{\theta} = (W_1, \beta_1 + \delta_1, W_2, \beta_2 + \delta_2, \ldots, W_L, \beta_L + \delta_L)$, where the noise terms are mutually independent and identically distributed zero-mean Gaussian, $\delta_l \sim N(0, \sigma I_{n_l})$, with some fixed standard deviation $\sigma > 0$. We denote the bias with the noise term as $b_l = \beta_l + \delta_l$. For this random variable, we consider the expected volume of the non-linear locus $\mathcal{B}_{\mathcal{N}_{\tilde{\theta}}}$ around any input point $x$ and define a corresponding density as the limit:

$$f(x) = \lim_{\epsilon \to 0} \frac{1}{Z_\epsilon} \underset{\tilde{\theta}}{\mathbb{E}} \left[ \mathrm{vol}_{n_0 - 1}(\mathcal{B}_{\mathcal{N}_{\tilde{\theta}}} \cap B_\epsilon(x)) \right], \quad x \in \mathbb{R}^{n_0}. \tag{3}$$

Here the expectation is taken with respect to the random parameter $\tilde{\theta}$ or more specifically the noise terms $\delta_1, \ldots, \delta_l$. The limit is taken with respect to the radius $\epsilon$ of a ball $B_\epsilon(x)$ around the input point $x$, and the normalization factor is given by the volume of the ball:

$$Z_\epsilon = \mathrm{vol}_{n_0}(B_\epsilon(0)) \propto \epsilon^{n_0}. \tag{4}$$

We illustrate this definition in Figure 1, where we numerically estimate the density function $f$ over the input space for a network with two-dimensional input. We demonstrate the impact of $\sigma$ on the local complexity qualitatively in Appendix B.1. We now define the local complexity of our neural network as the expectation of $f$ over the input data distribution. We denote by $p$ the probability distribution of the data over the input space $\mathbb{R}^{n_0}$, which we will assume to have a density and a compact support $\Omega$.

**Definition 1** (Local Complexity). *We define the* local complexity *of a network $\mathcal{N}$ at parameter $\theta$ with respect to the input data distribution $p$ as*

$$\mathbf{LC}(\mathcal{N}_\theta, p) = \underset{x \sim p}{\mathbb{E}} [f(x)]. \tag{5}$$

For simplicity of notation we will omit the arguments $\mathcal{N}_\theta$ and $p$ when there is no risk of confusion. We define local complexity by taking the expectation of $f$ over the data distribution to estimate the density of linear regions near the dataset, where model complexity is most relevant. To provide further intuition for this definition and later results, we conduct a direct computation of the local complexity for a few illustrative examples in Appendix A.2.

## 3.2 Towards a Theoretical Understanding of the Local Complexity

We can now introduce our first results in understanding this measure of the complexity of a neural network with respect to the data distribution $p$. As before, we denote the pre-bias value of the $i$th neuron of the network at input $x$ by $z_i(x)$, for $i = 1, \ldots, \sum_{\ell=1}^{L-1} n_\ell$. For a neuron $z_i$ in layer $l$, we say that $z_i$ *is good at $x$* if the computation graph of the network evaluated at input $x$ contains a path of active neurons $z_{j_{l+1}}, \ldots, z_{j_{L-1}}$ from layers $l+1$ to $L$, where for each neuron in this path, $z_{j_i}(x) > b_{j_i}$. In particular, this means that the neuron $z_i$ affects the network's output when evaluated at $x$. More details on this can be found in Appendix A.1. We denote by $\rho_{b_i}$ the Gaussian density function for the bias of neuron $z_i$ perturbed by $\delta_i$. We will denote by $\nabla z_i(x)$ the gradient of function $z_i$ with respect to $x$ where this is well defined. The non-differentiable points form a null set and are inconsequential in the following results. The following theorem provides an explicit formula for computing the local complexity. A proof of this theorem can be found in Appendix A.3.

**Theorem 2.** *Let $\rho_{b_i}(x) = N(\beta_i, \sigma)$ be the density for the bias of neuron $z_i$. Then the following holds:*

$$\mathbf{LC} = \sum_{neuron\ z_i} \underset{x, \tilde{\theta}}{\mathbb{E}} \left[ \|\nabla z_i(x)\|_2\, \rho_{b_i}(z_i(x))\, \mathbb{1}_{z_i\ is\ good\ at\ x} \right], \tag{6}$$

*where for each neuron the expectation is taken over $\tilde{\theta}$ and $x \sim p$.*

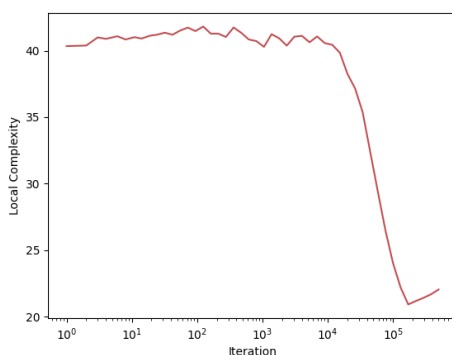 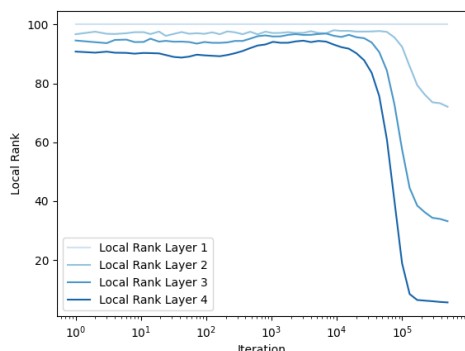

Figure 2: Relation between the local rank (10) at intermediate layers and the local complexity (2) of linear regions in the terminal phase of training. We train an 4 layer MLP with 200 neurons per layer to estimate the learn a map between two multivariate Gaussians with a random cross-covariance matrix. More information can be found in Appendix B.2.

This theorem gives us a way to compute the local complexity empirically by computing the gradients at each neuron and estimating (6) using samples, which we leverage in our numerical experiments in Figures 1, 2, 3. We can now introduce bounds on the local complexity, which will be useful for our later analysis because they allow us to focus on the gradient terms.

**Corollary 3.** *In the same setting as Theorem 2, let $C_{grad}$ be an upper bound on the norm of the gradient of every neuron $z_i$, $\|\nabla z_i(x)\| \leq C_{grad}$ for all $x \in \Omega$, $\tilde{\theta} = (W_1, \beta + \delta_1, \ldots, W_L, \beta + \delta_L)$, let $C_{bias} = \frac{1}{\sqrt{2\pi}\sigma}$, and let $B = \mathbb{E}_{\tilde{\theta}, x \sim p} \left[ \sum_{neuron\, z_i} \mathbb{1}_{z_i\, not\, good\, at\, x} \right]$ denote the expected number neurons that are not good. Then we have that:*

$$\textbf{LC} \leq C_{bias} \sum_{neuron\, z_i} \mathbb{E}_{\tilde{\theta}; x \sim p} \left[ \|\nabla z_i(x)\|_2 \right]. \tag{7}$$

*Furthermore, for any $\eta > 0$ there are constants $c_{bias}^\eta = \frac{1}{\sqrt{2\pi}\sigma} e^{\frac{-\eta^2}{2\sigma^2}}$ and $\bar{\xi}_\eta = \Theta \left( e^{\frac{-\eta^2}{2\sigma^2}} / \eta^2 \right)$[1] such that:*

$$\textbf{LC} \geq c_{bias}^\eta \sum_{neuron\, z_i} \mathbb{E}_{\tilde{\theta}; x \sim p} \left[ \|\nabla z_i(x)\|_2 \right] - \bar{\xi}_\eta - B \cdot C_{grad} \cdot C_{bias}. \tag{8}$$

We note that the term $B \cdot C_{grad} \cdot C_{bias}$, while a necessary inclusion based on our proof technique, may be quite small. Indeed at initialization Hanin & Rolnick (2019a, Appendix D) observe that for any neuron $z$ and $x \in \mathbb{R}^{n_0}$, $\mathbb{P}(z\text{ is good at }x) \geq 1 - \sum_{l=1}^L 2^{-n_l}$. Thus, at initialization, $B \leq N L \, 2^{-n}$, where $N = \sum_l n_l$ and $n = \min_l n_l$, which decays exponentially with the width. Our empirical results in Appendix B.4 have shown that, for fully connected networks of reasonable width, $B$ is typically measured to be constant at $0$. Similarly, the term $\bar{\xi}_\eta$ can also be small, and notably is asymptotically smaller than $c_{bias}^\eta$ for large values of $\eta$.

## 4 CONNECTIONS TO THE RANK OF LEARNED REPRESENTATIONS

We define the feature manifold at layer $l$, denoted $\mathcal{M}_l \subseteq \mathbb{R}^{n_l}$, to be the pre-activation values of the $l$th layer when evaluated on the support of the input data distribution, $\Omega$. We proceed by introducing a measure of the dimension of the feature manifold, which we call the *local rank*. We write $\mathbf{z}_l = [z_{l,1}, \ldots, z_{l,n_l}]$ for the vector of pre-bias pre-activations at the $l$th layer and $J_x \mathbf{z}_l(x) = [\nabla_x z_{l,1}(x), \ldots, \nabla_x z_{l,n_l}(x)]^T$ for the Jacobian with respect to the input. With this notation, we can write the feature manifold as $\mathcal{M}_l = \mathbf{z}_l(\Omega)$.

---

[1] We use the standard Big Theta notation $f(x) = \Theta(g(x))$ to signify that there exist $c_1, c_2, x_0$ such that $c_1 g(x) \leq f(x) \leq c_2 g(x)$ for all $x > x_0$.

**Definition 4** (Local Rank). *We define the* local rank *(LR) of the lth layer's features as the expectation value of the dimension of the feature manifold over the input data distribution:*

$$\mathbf{LR}_l = \mathop{\mathbb{E}}_{x \sim p, \tilde{\theta}} \left[ \mathrm{rank}(J_x \mathbf{z}_l(x)) \right]. \tag{9}$$

*We will find it more convenient, both numerically and analytically, to work with the approximate rank,* $\mathrm{rank}_\epsilon(A) = |\{\sigma \geq \epsilon \colon \sigma \text{ singular value of } A\}|$*, which satisfies* $\lim_{\epsilon \to 0} \mathbf{LR}_l^\epsilon = \mathbf{LR}_l$,

$$\mathbf{LR}_l^\epsilon = \mathop{\mathbb{E}}_{x \sim p, \tilde{\theta}} \left[ \mathrm{rank}_\epsilon(J_x \mathbf{z}_l(x)) \right]. \tag{10}$$

Note that a generic input point $x$ will lie in the interior of a linear region of $\mathbf{z}_l$. The Jacobian matrix $J_x \mathbf{z}_l$ provides the linearization of $\mathbf{z}_l$ over the interior of of each linear region. The null space of the Jacobian indicates the dimensions of the input space that are discarded in the computation of the outputs near the input $x$, and the rank is equal to the dimension of the set of feature values traced as we perturb the input $x$. Thus, this is a meaningful measure of the rank of the learned representations.

Our aim is to provide a connection between the local rank of the representations in Definition 4 and the local complexity of the learned functions in Definition 1. The following result, proven in Appendix A.5, links the local rank and the local complexity:

**Theorem 5.** *For any $\epsilon > 0$, the local ranks across layers can be bounded in terms of the local complexity as follows:*

$$\frac{1}{n_0 \, C_{bias}} \mathbf{LC} \leq \sum_{l=1}^{L} \sqrt{C_{grad}^2 \, \mathbf{LR}_l^\epsilon + \epsilon^2 n_l}. \tag{11}$$

*Moreover, in the same setting as in Corollary 3,*

$$\sum_{l=1}^{L} \mathbf{LR}_l^\epsilon \leq \frac{1}{c_{bias}^\eta \epsilon^2} \left[ \mathbf{LC} + \bar{\xi}_\eta + B \cdot C_{grad} \cdot C_{bias} \right]. \tag{12}$$

We can also make a weaker claim about the exact local rank, which we prove in Corollary 14 in the appendix: $\mathbf{LC} \leq n_0 \, C_{\text{bias}} C_{\text{grad}} \sum_{l=1}^{L} \sqrt{\mathbf{LR}_l}$.

We showcase the relation between the local rank and local complexity in a simple example. Figure 2 shows the evolution of $\mathbf{LC}$ and $\mathbf{LR}$ during training, for Gaussian input and output data. In this example both quantities appear to be tightly related and we observe a stark and sudden drop in the local rank late in training. The information theoretic properties of the rank of representations for this particular example has been studied in the prior work of Patel & Shwartz-Ziv (2024). While this behavior is not unique to this example, on other datasets the dynamics of the local rank can become much more complex and it is not yet fully understood, as we showcase in Figure 12 in the appendix.

## 5    NETWORKS WITH LOWER LOCAL COMPLEXITY MAY BE MORE ROBUST

Neural networks have been shown to sometimes converge to solutions that exhibit neural collapse (Papyan et al., 2020). In this case, the networks have a low within-class variance of representations in the last hidden layer, implying that the learned function is flat around the data points. We will attempt to understand this specific geometric property by considering the total variation of a trained neural network over the data distribution, which we define as $\mathbf{TV} = \mathbb{E}_{\tilde{\theta}, x \sim p}[\|\nabla_x \mathcal{N}(x)\|]$. A low expected total variation indicates that the gradient of the network function is typically small over the data distribution. Consequently, these networks develop stable regions around training data points where the function is nearly constant, aligning with the characteristics of neural collapse.

We remark that low total variation has implications for adversarial robustness. Standard methods for generating adversarial examples, such as Projected Gradient Descent (PGD), rely on first-order optimization techniques for constructing adversarial examples (Madry et al., 2018). Low total variation with respect to the data distribution makes it harder for such methods to find adversarial examples, since small gradients limit the effectiveness of first-order optimization. In some settings, the total variation can be related directly to the existence of adversarial examples. Suppose we have a univariate network $\mathcal{N}_\theta \colon \mathbb{R} \to \mathbb{R}$ for classification, with a decision boundary at $\mathcal{N}_\theta(x) = 0$. For a given $\bar{x}$ with $\mathcal{N}_\theta(\bar{x}) > 0$, a point $x \in B_\epsilon(\bar{x})$ is an adversarial example if $\mathcal{N}_\theta(x) < 0$. Then we have the following proposition, which we prove in Appendix A.7.

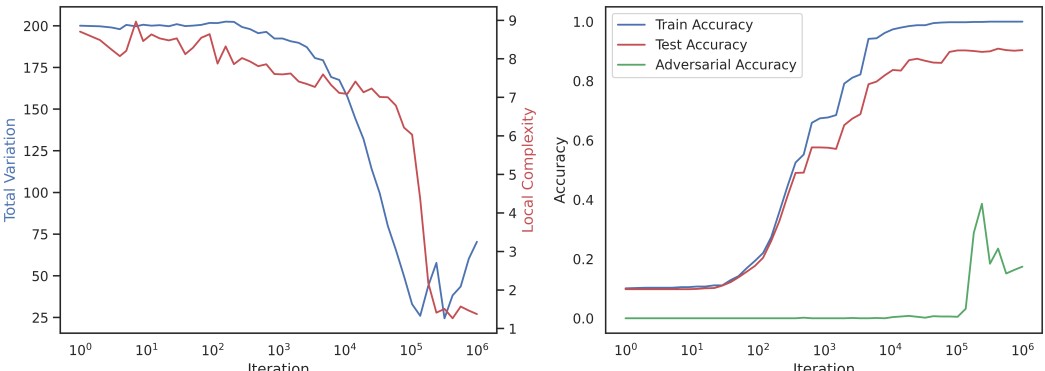

Figure 3: Drop in the expected total variation (7) and local complexity (2) of a network in the terminal phase of training. We find that this corresponds to an increase in the adversarial robustness of our network. Here we train a 4 layer MLP with 200 neurons in each layer on a subset of 1000 images from the MNIST dataset. We use an initialization scale that is 2x the standard He initialization. More information and an ablation on the initialization scale is available in Appendix B.3.

**Proposition 6.** *Suppose our data distribution admits a density function $p$ with support $\Omega$. Consider a point $\bar{x}$ in the interior of $\Omega$, with classification margin $\mathcal{N}_\theta(\bar{x}) > \gamma$. For any $\epsilon > 0$ with $B_\epsilon(\bar{x}) \in \Omega$, let $c_\epsilon = \inf_{x \in B_\epsilon(\bar{x})} p(x)$. Then $\mathbf{TV} \le c_\epsilon \gamma$ implies there are no adversarial examples in $B_\epsilon(\bar{x})$.*

Empirical results have shown that a drop in the local number of linear regions is accompanied by an increase in adversarial robustness (Humayun et al., 2024b). We can understand this by developing a bound between the local complexity and the expected total variation of that network. A full proof can be found in Appendix A.8.

**Theorem 7.** *Let $g_l$ denote the rest of the network after the $l$th layer, so that $\mathcal{N}_\theta = g_l \circ \phi(\mathbf{z}_l - b_l)$. Let $C_l$ denote the Lipschitz constant of $g_l$. Then with the same setting and notations as Theorem 2:*

$$\mathbf{TV} \cdot \frac{Lc_{bias}^\eta}{\max_{1 \le l \le L} C_l} - \bar{\xi}_\eta - B \cdot C_{grad} \cdot C_{bias} \le \mathbf{LC} . \tag{13}$$

This bound could help explain the findings of Humayun et al. (2024b), where ReLU networks trained in a classification task converged to solutions that are flat near data points and the non-linear locus is concentrated around the decision boundaries. We empirically demonstrate this behavior on the MNIST dataset in Figure 3. We note however that this theoretical result may not fully explain the relationship between the total variation and the local complexity during training. Indeed, in Appendix B.3 we illustrate that the dynamics can be more complex in general. We can view this as a consequence of the relationship between $\mathbf{TV}$ and the Lipschitz constants $\max_{1 \le l \le L} C_l$. A more detailed empirical study analysis on the tightness of this bound can be found in Appendix B.4.

## 6 THE DROP OF LOCAL COMPLEXITY, AND A CONNECTION TO GROKKING

In this section, we explore ways in which the local complexity might be implicitly minimized during training via representation cost and implicit regularization of weight matrices.

### 6.1 REPRESENTATION COST

The representation cost of a function $f$ is the smallest possible parameter norm needed for a neural network $\mathcal{N}_\theta$ to exactly compute $f$. We define this as $R(f) = \inf_\theta \{\|\theta\|_F : \mathcal{N}_\theta(x) = f(x) \text{ for all } x \in \Omega\}$.[2] Prior works have analyzed the representation cost of shallow networks of arbitrary width (Savarese et al., 2019). The representation cost for linear networks can be explicitly calculated in certain cases, and is often connected to sparsity. For instance, in fully-connected linear networks, it

---

[2]One may take $\inf_\theta \{\|\theta\|_F : \|f - \mathcal{N}_\theta\| \le \epsilon\}$ for some appropriate norm and a limit in $\epsilon$.

is a Schatten quasi norm of the end-to-end matrix (Dai et al., 2021). Deeper nonlinear networks also share a connection between the representation cost and sparsity in terms of rank (Jacot, 2023a). The following proposition, which we prove in Appendix A.9, provides a way to view the representation cost as a metric of sparsity, this time in terms of the linear regions.

**Proposition 8.** *In the same setting as Theorem 2, where $n_l$ is the maximum hidden layer dimension,*

$$\frac{n_0}{C_{bias}} \mathbf{LC} \leq n_l^{\frac{1-L}{2}} L^{1-\frac{L}{2}} R(\mathcal{N}_\theta)^L. \tag{14}$$

This bound provides some understanding of how weight decay and the resulting reduction of parameter norms may play a role in the simplification of linear regions that we find late in training, as we will discuss below in Corollary 10.

## 6.2 LINKING LOCAL COMPLEXITY TO NEURAL NETWORK OPTIMIZATION

Humayun et al. (2024b) presents empirical results that relate grokking to a migration of the linear regions in the terminal phase of training. In particular, they find a drop in the local number of linear regions near data points late in training. We aim to understand this drop of linear regions late in training as a drop in the local complexity. We can leverage as a heuristic the view of grokking provided by Lyu et al. (2024), who show grokking can be induced by a dichotomy of early and late phase implicit biases. This is only a heuristic way for us to view grokking in our setting, since that work requires the network to be *everywhere* $\mathcal{C}^2$ smooth, which is only true *almost* everywhere for our ReLU networks. However, it may be possible to generalize their result with a careful analysis with Clarke sub-differentials (Ji & Telgarsky, 2020; Clarke, 1975). Nevertheless, following Lyu et al. (2024) we consider:

$$\frac{d\theta}{dt} = -\nabla\mathcal{L}(\theta) - \lambda\|\theta\|_2.$$

They show that networks will first operate in the "kernel" regime (Jacot et al., 2018), during which the parameters do not move far from initialization. We show in Appendix A.10 that this implies that the local complexity also does not change much from initialization in shallow networks. After enough time, the network will eventually enter the "rich" regime and converge in direction to a KKT point of the following optimization problem; where $\{(x_i, y_i)\}_{i=1}^n \subseteq \mathbb{R}^{n_0} \times \{-1, 1\}$ is the training dataset:

$$\min_\theta \frac{1}{2}\|\theta\|^2 \quad \text{s.t.} \quad y_i\mathcal{N}_\theta(x_i) \geq 1, \ \forall i \in [n]. \tag{15}$$

In this setting, Timor et al. (2023) show that the global optimum of (15) has bounded ratios between the Frobenius norm and operator norm of weight matrices. We can relate this to the local complexity and show that in this setting the local complexity is also bounded as follows.

**Proposition 9.** *Let $\{(x_i, y_i)\}_{i=1}^n \subseteq \mathbb{R}^{n_0} \times \{-1, 1\}$ be a binary classification dataset, and assume that there is $i \in [n]$ with $\|x_i\| \leq 1$. Assume that there is a fully-connected neural network $\mathcal{N}$ of width $m \geq 2$ and depth $k \geq 2$, such that for all $i \in [n]$ we have $y_i\mathcal{N}(x_i) \geq 1$, and the weight matrices $W_1, \ldots, W_k$ of $\mathcal{N}$ satisfy $\|W_i\|_F \leq B$ for some $B > 0$. Let $\mathcal{N}_\theta$ be a fully-connected neural network of width $m' \geq m$ and depth $k' > k$ parameterized by $\theta$. Let $\theta^* = [W_1^*, \ldots, W_L^*]$ be a global optimum of the above optimization problem (15). Then, assuming the same setting as Theorem 2, we have the following bound on the local complexity:*

$$\frac{1}{L\max_{l\in[L]}\|W_i^*\|_{op}} \left(\frac{n_0}{C_{bias}\, n_l^{\frac{1-L}{2}} L^{1-\frac{L}{2}}} \mathbf{LC}\right)^{\frac{1}{L}} - \gamma \leq \sqrt{2} \cdot \left(\frac{B}{\sqrt{2}}\right)^{\frac{k}{L}} \cdot \sqrt{\frac{L+1}{L}}, \tag{16}$$

*where, $\gamma = \|W_i^*\|_F \left(\sqrt{\frac{1}{\|W_l^*\|_{op}}} - \sqrt{\frac{1}{\|W_i^*\|_{op}}}\right)^2$.*

We observe that the result in Proposition 9 is rigorous, but the corresponding bound only holds when our network is at the global minimum of (15). Another view we can take is by considering the norm of the weights. Lyu et al. (2024) show in the rich phase of training that $\|\theta(t)\|_2 = \Theta\left((\log\frac{1}{\lambda})^{1/L}\right)$. If we assume that this holds, using calculations in Appendix A.9, we can show that the local complexity is asymptotically bounded by the weight decay parameter $\lambda$.

**Corollary 10.** *[Informal] Suppose that* $\|\theta(t)\|_2 = \Theta\left((\log\frac{1}{\lambda})^{1/L}\right)$ *holds. Then, in the "rich" phase of training the local complexity is bounded:*

$$\frac{n_0}{C_{bias}n_l^{\frac{1-L}{2}}L^{1-\frac{L}{2}}}\, \mathbf{LC} \le \Theta(\log\tfrac{1}{\lambda}). \tag{17}$$

We empirically validate this claim in Figure 13, where we demonstrate that the local complexity will typically be lower for networks trained with larger weight decay values. However, it should be noted that the bound in (8) does not leverage the dependence on the input data point, so it is likely that these bounds are loose, and could be improved through a more exact analysis.

## 7 CONCLUSIONS AND FUTURE WORK

**Summary**  We presented a framework for analyzing the distribution of linear regions of the functions parametrized by neural networks with piecewise linear activations. We introduced a measure of local complexity that is robust with respect to perturbations of the parameters and used this to gain insights into relevant aspects of learning such as robustness and representation learning. Specifically, we establish that networks that learn low-dimensional representations tend to exhibit a lower local complexity. Further, we connected the local complexity of linear regions to the total variation of the network functions and thus to robustness. We also analyze how the local complexity can be implicitly minimized during training by connecting it to properties of the weight matrices. Overall, this work contributes a theoretical framework relating geometric properties of ReLU networks, specifically the linear regions, to different aspects of learning, and illustrates interesting interrelations that we hope might motivate further investigations in this direction.

**Limitations and future research**  We focused on the ReLU activation function. We think that our proof techniques could be adapted to obtain results for more general piecewise linear activation functions. Such generalizations could be approached in a similar way as Tseran & Montúfar (2021) approached the analysis of expected complexity for maxout networks. Though we find interesting results suggesting the proposed local complexity measure might be implicitly minimized during training, a detailed analysis addressing the training dynamics of the local complexity remains an open problem for further research. Empirically, in certain settings we can often observe complex interactions between local complexity and local rank, as well as between local complexity and total variation. This suggests that the explicit relationship between the local complexity and other measures of model complexity may be much richer than what is covered by our theoretical results. Another natural direction would be to construct explicit bounds on the generalization gap based on the local complexity, as one would expect that networks with a simple structure in terms of their linear regions would also generalize well.

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

## A  Main Theoretical Results

### A.1  Notation and Setup

Let $\mathcal{N}$ be a fully connected network with $L$ layers with input dimension $n_0$ output dimension 1 and ReLU non-linearity function $\phi(x) = \max\{0, x\}$. We denote $\phi(\mathbf{v})$ for $v \in \mathbb{R}^n$ to be the ReLU function applied element-wise. For simplicity, we will typically make the assumption that $n_l = n_j$ for all hidden layers $j \in [L]$. We denote by $h_l$ the post-activations after layer $l$. That is,

$$\mathcal{N}(x) = W_L h_L(x).$$

Here, each $h_i : \mathbb{R}^{n_0} \to \mathbb{R}^{n_i}$ is of the form:

$$h_i(x) = \phi(W_i h_{i-1}(x) - \beta_i),$$

$$h_0(x) = x.$$

We will write the pre-activations at neuron $j$ as $z_j(x) = W_i^j h_{i-1}(x)$. We can then write the vector of pre-activations at layer $l$ as $\mathbf{z}_l = (z_{l,1}, z_{l,2}, \ldots, z_{l,n_l})$. We can then write

$$h_l(x) = \phi(\mathbf{z}_l(x) - \beta_l).$$

We also use $\beta_i^j$ denotes the $j$-th element of vector $\beta_i$. When it is clear, we will write $\beta_z = \beta_i^j$. If neurons are indexed by $i$, such as $z_i$, we can write $\beta_i = \beta_{z_i}$ to denote the bias associated to neuron $z_i$. We write as $l(z)$ to denote the layer index that neuron $z$ appears.

We will typically use $\beta_i$ to refer to the deterministic choice of biases, and reserve $b_i = \beta_i + \delta_i$ to refer to the random variable representing the biases plus noise. We denote by $\theta = [W_L, \ldots, W_l, \beta_L, \ldots, \beta_1]$ the parameters of a network. We will write $\mathcal{N}_\theta$ to denote the network $\mathcal{N}$ parameterized by $\theta$. We represent the random variable for our parameters as $\tilde{\theta} = [W_L, \ldots, W_l, b_L, \ldots, b_1]$. When $\tilde{\theta}$ is treated as a random variable, we also consider $\mathcal{N}_{\tilde{\theta}}(x)$ to be a random variable, along with the corresponding quantity $z_i(x)$, which represents the random variable associated with a neuron.

Now define:

$$S_z = \{x \in \mathbb{R}^{n_0} \mid z(x) - b_z = 0\},$$

as the set of points where neuron $z$ switchs from on to off. Furthermore, define

$$\mathcal{O} = \{x \in \mathbb{R}^{n_0} \mid \forall j \in [L] \; \exists \text{ neuron } z \text{ with } l(z) = j : \; \phi'(z(x) - b_z) \neq 0\},$$

$$\tilde{S}_z = S_z \cap \mathcal{O}.$$

Then, $\mathcal{O}$ is the set of inputs $x$ for which there exists an open path from $x$ to the output of the function $\mathcal{N}$. Thus, we can read $\tilde{S}_z$ as the collection of points in the input space where $z$ switches between its linear regions, and this appears in the function computed by $\mathcal{N}$. Notice also in the case of the ReLU activation function, we can re-write $\mathcal{O}$ as the following:

$$\mathcal{O} = \{x \in \mathbb{R}^{n_0} \mid \forall j \in [l] \; \exists \text{ neuron } z \text{ with } l(z) = j : \; z(x) - b_z \geq 0\}.$$

We will also define $\mathcal{B}_{\mathcal{N}}$ to be:

$$\mathcal{B}_{\mathcal{N}} = \{x \in \mathbb{R}^{n_0} : \nabla_x \mathcal{N}(\cdot) \text{ discontinuous at } x\},$$

which is the set of non-linearities of the function $\mathcal{N}$. We call this the nonlinear locus of $\mathcal{N}$.

### On a neuron being "good"

We will sometimes take a path-wise representation of a ReLU network. In this case, we will first write $z^{(l)}$ to denote a neuron in the $l$-th layer. Let $\gamma = (\gamma_1, \gamma_2, \ldots, \gamma_L)$ denote a path in the computation graph of $\mathcal{N}$, where each $\gamma_i$ indexes a neuron in the $i$-th layer. To clarify notation since the $i$-th neuron in a path will always be in the $i$-th layer, we will write $z_{\gamma_i} = z_\gamma^{(i)}$. We can also note that there is an associated sequence of weights on the edges of that computation graph, which we can denote by $w_\gamma^{(l)} =$ "weight connecting $z_\gamma^{(l-1)}$ to $z_\gamma^{(l)}$". More formally, if $W$ is the $l - 1$ layer weight matrix,

then $w_\gamma^{(l)} = W_{\gamma_{l-1}, \gamma_l}$. Denote by $\Gamma_i$ the set of all paths in the computation graph of $\mathcal{N}$ leading from the $i$-th input to the output node. We can now give a path-wise representation of our neural network $\mathcal{N}$ as:

$$\mathcal{N}(x) = \sum_{i=1}^{n_0} x_i \sum_{\gamma \in \Gamma_i} \prod_{l=1}^{L} \mathbb{1}_{\{z_\gamma^{(l)}(x) - b_z \geq 0\}} w_\gamma^{(l)} + \mathcal{N}(0).$$

In this case, a neuron in $\gamma$ is open when $z_\gamma^{(l)}(x) - b_z \geq 0$. A neuron $z$ is good at $x$ it is contained in a path $\gamma$ leading from the input to the output, where every neuron after $z$ is open.

## A.2 ILLUSTRATIVE EXAMPLES OF THE LOCAL COMPLEXITY

### COMPUTING THE LOCAL COMPLEXITY OF A SINGLE NEURON

As an illustrative example, and to gain some intuition for Theorem 2, we compute explicitly the local complexity of a single neuron. Our model is as follows, where $v, w, \beta \in \mathbb{R}$, and $\phi$ denotes the ReLU function. Our parameters are $\theta = (v, w, \beta)$, and our model is

$$\mathcal{N}_\theta(x) = v\phi(wx - \beta), \quad x \in \mathbb{R}.$$

Notice first that the breakpoint (non-linearity) of this function is always at $x = \frac{\beta}{w}$. Now recall the definition of the local complexity density function $f$:

$$f(x) = \lim_{\epsilon \to 0} \frac{1}{Z_\epsilon} \mathbb{E}_{\tilde{\theta}} \left[ \text{vol}_{n_0 - 1}(B_{\mathcal{N}_{\tilde{\theta}}} \cap B_\epsilon(x)) \right], \quad x \in \mathbb{R}^{n_0}.$$

For our setting here, $\tilde{\theta} = (w, b)$ where $b$ is Gaussian with variance $\sigma^2$ centered at $\beta$. The normalizaiton factor is given by $Z_\epsilon = 2\epsilon$ For now consider fixing $\epsilon > 0$, then notice that:

$$\mathbb{E}_{\tilde{\theta}} \left[ \text{vol}_{n_0 - 1}(B_{\mathcal{N}_{\tilde{\theta}}} \cap B_\epsilon(x)) \right] = \mathbb{E}_{b}[\mathbb{1}_{\frac{b}{w} \in (x - \epsilon, x + \epsilon)}]$$

$$= \mathbb{P}(\frac{b}{w} \in (x - \epsilon, x + \epsilon))$$

$$= \mathbb{P}(b \in (wx - w\epsilon, wx + w\epsilon))$$

$$= \int_{wx - w\epsilon}^{wx + w\epsilon} \rho_b(b) db$$

$$= \int_{x - \epsilon}^{x + \epsilon} |w| \rho_b(w\tilde{b}) d\tilde{b}.$$

Notice we gain a factor of $w$ in the integrand through a change of variables. We illustrate this because this is very similar to how the term $\nabla z(x)$ shows up in the proof of Theorem 2. In particular, this is one way to see how the co-area formula which we utilize in the main proof is a generalization of the typical change of variables formula. We can proceed now to see that:

$$f(x) = \lim_{\epsilon \to 0} \frac{1}{2\epsilon} \int_{x - \epsilon}^{x + \epsilon} |w| \rho_b(w\tilde{b}) d\tilde{b} = |w| \rho_b(wx).$$

Our local complexity for this single neuron is then given as follows, where $p$ is the data distribution:

$$LC = \mathbb{E}_{x \sim p} [f(x)] = \mathbb{E}_{x \sim p} [|w| \rho_b(wx)].$$

Notice this is precisely what we would arrive at by a direct application of Theorem 2 to our model.

### COMPUTING THE LOCAL COMPLEXITY OF A 2 HIDDEN LAYER NETWORK

To illustrate how these results start to generalize to deeper networks, we show a direct computation of the local complexity for a univariate neural network with one neuron in the first hidden layer and

one neuron in the second hidden layer. In particular, consider the following network parameterized by $\theta = (w_2, w_1, \beta_2, \beta_1)$,

$$\mathcal{N}(x) = \phi(w_2 \phi(w_1 x - \beta_1) - \beta_2).$$

We also will write,

$$z_1(x) = w_1 x,$$

and

$$z_2(x) = w_2 \phi(w_1 x - \beta_1) = w_2 \phi(z_1(x) - \beta_1).$$

Then computing derivatives on $\mathcal{N}$ gives that:

$$\mathcal{N}'(x) = \mathbb{1}_{z_2(x) > \beta_2} \mathbb{1}_{z_1(x) > \beta_1} w_1 w_2$$

The breakpoints at which $\mathcal{N}'$ is not continuous are then given by these indicator functions, so then:

$$
\begin{aligned}
\mathcal{B}_{\mathcal{N}} &= \{x : \mathcal{N}'(x) \text{ not continuous at } x\} \\
&= \{\frac{b_1}{w_1} \text{ if } z_2 \text{ open at } x = \frac{b_1}{w_1}\} \cup \{\frac{b_2}{w_1 w_2} + \frac{b_1}{w\ 1} \text{ if } z_1 \text{ open at } x = \frac{(b_2 + w_2 b_1)}{w_2 w_1}\}.
\end{aligned}
$$

Now let $\epsilon > 0$, and suppose that $b_1$ is normal with mean $\beta_1$ and variance $\sigma^2$ and that $b_2$ is normal with mean $\beta_2$ and variance $\sigma^2$. Then we have that

$$
\begin{aligned}
\mathbb{E}_{b_1, b_2} \left[\text{vol}_0(\mathcal{B}_{\mathcal{N}_{\hat{\theta}}} \cap (x - \epsilon, x + \epsilon)\right] &= \mathbb{E}_{b_1, b_2} \left[\mathbb{1}_{\frac{b_1}{w_1} \in (x - \epsilon, x + \epsilon)} \mathbb{1}_{z_2(\frac{b_1}{w_1}) > b_2}\right. \\
&\quad + \left. \mathbb{1}_{\frac{b_2}{w_1 w_2} + \frac{b_1}{w\ 1} \in (x - \epsilon, x + \epsilon)} \mathbb{1}_{z_1(\frac{b_2}{w_1 w_2} + \frac{b_1}{w\ 1}) > b_1}\right] \\
&= \mathbb{E}_{b_2} \left[\mathbb{E}_{b_1} \left[\mathbb{1}_{\frac{b_1}{w_1} \in (x - \epsilon, x + \epsilon)} \mathbb{1}_{z_2(\frac{b_1}{w_1}) > b_2}\right]\right] \\
&\quad + \mathbb{E}_{b_1} \left[\mathbb{E}_{b_2} \left[\mathbb{1}_{\frac{b_2}{w_1 w_2} + \frac{b_1}{w_1} \in (x - \epsilon, x + \epsilon)} \mathbb{1}_{z_1(\frac{b_2}{w_1 w_2} + \frac{b_1}{w\ 1}) > b_1}\right]\right].
\end{aligned}
$$

Now first we compute on the first term,

$$
\begin{aligned}
\mathbb{E}_{b_1} \left[\mathbb{1}_{\frac{b_1}{w_1} \in (x - \epsilon, x + \epsilon)} \mathbb{1}_{z_2(\frac{b_1}{w_1}) > b_2}\right] &= \int_{-\infty}^{\infty} \rho_{b_1}(b) \mathbb{1}_{\frac{b_1}{w_1} \in (x - \epsilon, x + \epsilon)} \mathbb{1}_{z_2(\frac{b}{w_1}) > b_2} db \\
&= \int_{w_1(x - \epsilon)}^{w_1(x + \epsilon)} \rho_{b_1}(b) \mathbb{1}_{z_2(\frac{b}{w_1}) > b_2} db \\
&= \int_{(x - \epsilon)}^{(x + \epsilon)} |w_1| \rho_{b_1}(w_1 b) \mathbb{1}_{z_2(b) > b_2} db.
\end{aligned}
$$

So then,

$$\mathbb{E}_{b_2} \left[\mathbb{E}_{b_1} \left[\mathbb{1}_{\frac{b_1}{w_1} \in (x - \epsilon, x + \epsilon)} \mathbb{1}_{z_2(\frac{b_1}{w_1}) > b_2}\right]\right] = \mathbb{E}_{b_2} \left[\int_{(x - \epsilon)}^{(x + \epsilon)} |w_1| \rho_{b_1}(w_1 b) \mathbb{1}_{z_2(b) > b_2} db\right].$$

From the above we can see that by taking limits we get:

$$\lim_{\epsilon \to 0} \frac{1}{Z_\epsilon} \mathbb{E}_{b_2} \left[\mathbb{E}_{b_1} \left[\mathbb{1}_{\frac{b_1}{w_1} \in (x - \epsilon, x + \epsilon)} \mathbb{1}_{z_2(\frac{b_1}{w_1}) > b_2}\right]\right] = \mathbb{E}_{b_2} \left[|w_1| \rho_{b_1}(w_1 x) \mathbb{1}_{z_2(x) > b_2} db\right].$$

Now on the other term we calculate:

$$\mathop{\mathbb{E}}_{b_2}\big[\mathbb{1}_{\frac{b_2}{w_1 w_2}+\frac{b_1}{w_1}\in(x-\epsilon,x+\epsilon)}\mathbb{1}_{z_1(\frac{b_2}{w_1 w_2}+\frac{b_1}{w_1})>b_1}\big]$$

$$=\int_{-\infty}^{\infty}\rho_{b_2}(b)\mathbb{1}_{\frac{b}{w_1 w_2}+\frac{b_1}{w_1}\in(x-\epsilon,x+\epsilon)}\mathbb{1}_{z_1(\frac{b}{w_1 w_2}+\frac{b_1}{w_1})>b_1}db$$

$$=\int_{-\infty}^{\infty}\rho_{b_2}(b)\mathbb{1}_{\frac{b}{w_1 w_2}\in(x-\epsilon-\frac{b_1}{w_1},x+\epsilon-\frac{b_1}{w_1})}\mathbb{1}_{z_1(\frac{b}{w_1 w_2}+\frac{b_1}{w_1})>b_1}db$$

$$=\int_{-\infty}^{\infty}\rho_{b_2}(b)\mathbb{1}_{b\in w_1 w_2(x-\epsilon-\frac{b_1}{w_1},x+\epsilon-\frac{b_1}{w_1})}\mathbb{1}_{z_1(\frac{b}{w_1 w_2}+\frac{b_1}{w_1})>b_1}db$$

$$=\int_{w_1 w_2(x-\epsilon-\frac{b_1}{w_1})}^{w_1 w_2(x+\epsilon-\frac{b_1}{w_1})}\rho_{b_2}(b)\mathbb{1}_{z_1(\frac{b}{w_1 w_2}+\frac{b_1}{w_1})>b_1}db$$

$$=\int_{(x-\epsilon-\frac{b_1}{w_1})}^{(x+\epsilon-\frac{b_1}{w_1})}|w_1 w_2|\rho_{b_2}(w_1 w_2 b)\mathbb{1}_{z_1(b+\frac{b_1}{w_1})>b_1}db.$$

From the above equation, we can take limits and see that:

$$\lim_{\epsilon\to0}\frac{1}{Z_\epsilon}\mathop{\mathbb{E}}_{b_1}\big[\mathop{\mathbb{E}}_{b_2}[\mathbb{1}_{\frac{b_2}{w_1 w_2}+\frac{b_1}{w_1}\in(x-\epsilon,x+\epsilon)}\mathbb{1}_{z_1(\frac{b_2}{w_1 w_2}+\frac{b_1}{w_1})>b_1}]\big]=\mathop{\mathbb{E}}_{b_1}[|w_1 w_2|\rho_{b_2}(w_1 w_2(x-\frac{b_1}{w_1}))\mathbb{1}_{z_1(x)>b_1}].$$

We can now see that $|z_1'(x)|=|w|$ and $|z_2'(x)|=\mathbb{1}_{z_1(x)>b_1}|w_1 w_2|$. Furthermore, $w_1 w_2(x-\frac{b_1}{w_1})=w_2(w_1 x-b_1)=z_2(x)$ on $\{z_1(x)>b_1\}$. Now notice that $z_2$ is always good at $x$ since it is directly connected to the output layer. So then,

$$f(x)=\mathop{\mathbb{E}}_{b_1}[|z_2'(x)|\rho_{b_2}(z_2(x))]+\mathop{\mathbb{E}}_{b_2}[|z_1'(x)|\rho_{b_1}(z_1(x))\mathbb{1}_{z_1\text{ good at }x}]$$

$$=\mathop{\mathbb{E}}_{b_1}[|z_2'(x)|\rho_{b_2}(z_2(x))\mathbb{1}_{z_2\text{ good at }x}]+\mathop{\mathbb{E}}_{b_2}[|z_1'(x)|\rho_{b_1}(z_1(x))\mathbb{1}_{z_1\text{ good at }x}].$$

Which, after taking expectations over $x\sim p$, is equivalent to the main result in Theorem 2.

## A.3 Proof of Theorem 2

The proof of this result will follow an argument that is closely inspired in the work of Hanin & Rolnick (2019a). Key to our proof is use of the generalized co-area formula, which we review here for completeness.

### A.3.1 Generalized Co-Area Formula

For $u$ with support on $\Omega\subseteq\mathbb{R}^n$, where $u:\mathbb{R}^n\to\mathbb{R}^k$ and is Lipschitz, for an $L^1$ function $g$, we have that:

$$\int_\Omega g(x)\|J_k u(x)\|dx=\int_{\mathbb{R}^k}\int_{u^{-1}(t)}g(x)d\text{vol}_{n-k}(x)dt.$$

Where:

$$\|J_k u(x)\|=\det(Ju(x)Ju(x)^T)^{\frac{1}{2}}.$$

### A.3.2 Lemma 11

The following lemma bears strong resemblance to Proposition 9 in the work of Hanin & Rolnick (2019a).

**Lemma 11.** *We have that almost surely:*

$$\mathcal{B}_\mathcal{N}=\bigcup_{z\text{ neuron}}\tilde{S}_z.$$

*Furthermore, this union is disjoint modulo a null set with respect to the Hausdorff $n_0-1$ measure.*

*Proof.* We will first check that $\mathcal{B}_{\mathcal{N}} \subseteq \bigcup_{z \text{ neuron}} \tilde{S}_z$ by checking if the following equation (18) holds:

$$\bigcap_{z \text{ neuron}} \tilde{S}_z^c \subseteq \mathcal{B}_{\mathcal{N}}^c. \tag{18}$$

Note that this suffices since, $\left( \bigcup_{z \text{ neuron}} \tilde{S}_z \right)^c = \bigcap_{z \text{ neuron}} S_z^c$. Fix $x \in \left( \bigcup_{z \text{ neuron}} \tilde{S}_z \right)^c$. We will now write:

$$Z_x^+ = \{z \text{ neurons} \,|\, z(x) - b_z > 0\},$$
$$Z_x^- = \{z \text{ neurons} \,|\, z(x) - b_z < 0\},$$
$$Z_x^0 = \{z \text{ neurons} \,|\, z(x) - b_z = 0\}.$$

Notice that on the left hand side of (18) we have a finite intersection of open sets which is also an open set. As a consequence, the map $x \to Z_x^*$ must be locally constant, and there exists some $\epsilon-$neighborhood around $x$ so that $\|x - y\| \leq \epsilon$ implies that:

$$Z_x^- \subseteq Z_y^-, \quad Z_x^+ \subseteq Z_y^+, \quad Z_y^+ \cup Z_y^0 \subseteq Z_x^+ \cup Z_x^0. \tag{19}$$

Now to prove (18) we will leverage the path-wise representation of our neural network $\mathcal{N}$, following the notation in Appendix A.1.

$$\mathcal{N}(y) = \sum_{i=1}^{n_0} y_i \sum_{\gamma \in \Gamma_i} \prod_{l=1}^{L} \mathbb{1}_{\{z_\gamma^{(l)}(y) - b_z \geq 0\}} w_\gamma^{(l)} + \mathcal{N}(0).$$

Now we have that, since $x \in \left( \bigcup_{z \text{ neuron}} \tilde{S}_z \right)^c$, for every path $\gamma$ that hits $z \in Z_x^0$:

$$\exists j \in [L] : z_\gamma^{(j)} \in Z_x^-.$$

By extension and by (19) we have that this holds in a neighborhood of $x$:

$$\forall y \in \mathbb{R}^{n_0} : \|x - y\| \leq \epsilon \implies z_\gamma^{(j)} \in Z_y^-.$$

And for $y$ in a neighborhood of $x$:

$$\mathcal{N}(y) = \sum_{i=1}^{n_0} y_i \sum_{\gamma \in \Gamma_i, \, \gamma \subseteq Z_x^+} \prod_{l=1}^{L} \mathbb{1}_{\{z_\gamma^{(l)}(y) - b_z \geq 0\}} w_\gamma^{(l)} + \mathcal{N}(0).$$

But then notice that we also have:

$$z(x) - b_z > 0 \implies z(y) - b_z > 0.$$

So then, for $y$ close to $x$,

$$\mathbb{1}_{\{z_\gamma^{(l)}(x) - b_z \geq 0\}} = \mathbb{1}_{\{z_\gamma^{(l)}(y) - b_z \geq 0\}},$$

and so we can write:

$$\mathcal{N}(y) = \sum_{i=1}^{n_0} y_i \sum_{\gamma \in \Gamma_i, \, \gamma \subseteq Z_x^+} \prod_{l=1}^{L} \mathbb{1}_{\{z_\gamma^{(l)}(x) - b_z \geq 0\}} w_\gamma^{(l)} + \mathcal{N}(0).$$

From which it is clear that $\partial \mathcal{N} / \partial y_i$ is independent of $y$. Therefore, the function $\mathcal{N}$ is a continuous linear function in a neighborhood of $x$ and we have shown (18). We will now aim to show:

$$\bigcup_{z \text{ neuron}} \tilde{S}_z \subseteq \mathcal{B}_{\mathcal{N}}. \tag{20}$$

First note that since our biases are admit a density with respect to the Lebesgue measure, we have that the following holds almost surely (a.s.) for $j \neq i$:

$$\text{vol}_{n_0-1}(S_{z_i} \cap S_{z_j}) = 0 \quad \text{(a.s.)}. \tag{21}$$

So then (20) would follow almost surely from showing that:

$$\bigcup_{z \text{ neuron}} \left( \tilde{S}_z \setminus \bigcup_{z' \neq z} S_{z'} \right) \subseteq \mathcal{B}_{\mathcal{N}}. \tag{22}$$

Now pick $x \in \left( \tilde{S}_z \setminus \bigcup_{z' \neq z} S_{z'} \right)$ for some fixed neuron $z$. Note that in a small enough $\epsilon-$neighborhood of $x$, we have that $y \to z(y)$ is linear in $y$. So then it follows that in this neighborhood of $x$, $\tilde{S}_z \setminus \bigcup_{z' \neq z} S_{z'}$ is a hyperplane of co-dimension 1. Pick $y_1$ so that $0 < \|x - y_1\| \leq \epsilon$ and $z(y_1) > b_z$ and $y_2$ so that $0 < \|x - y_2\| \leq \epsilon$ and $z(y_2) < b_z$. So then it follows that $x$ separate s two different activation patterns, and by assumption we have that $x$ is a discontinuity point of $\nabla_x \mathcal{N}(x)$. This proves equation (22).

Notice we have already proved that this union is (a.s.) almost everywhere disjoint with respect to the Hausdorff $n_0 - 1$ measure in equation (21). The claim follows. $\qquad\square$

### A.3.3 LEMMA 12

The following lemma is from Hanin & Rolnick (2019a) and is provided here with minor tweaks for convenience.

**Lemma 12.** *Let $z_1, \ldots, z_k$ be distinct neurons in the same layer of $\mathcal{N}$. Then for any compact $K \subset \mathbb{R}^{n_0}$,*

$$\mathbb{E}_{\tilde{\theta}}[vol_{n_0-k}(\tilde{S}_{z_1,\ldots,z_k} \cap K)] = \int_K \mathbb{E}_{\tilde{\theta}}[\|J_{z_1,\ldots,z_k}(x)\| \cdot \rho_{b_1,\cdots,b_k}(z_1(x), \ldots z_k(x)) \mathbb{1}_{\forall j: \ z_j \ good \ at \ x}]dx,$$

*where the expectation is taken with respect the noise terms $\delta_i$ in the biases.*

*Proof.* Let $z_1, \ldots, z_k$ be some distinct neurons in $\mathcal{N}$. Let $K \subseteq \mathbb{R}^{n_0}$. Then notice that:

$$\text{vol}_{n_0-k} \left( \tilde{S}_{z_1,\ldots,z_k} \cap K \right) = \int_{\tilde{S}_{z_1,\ldots,z_k} \cap K} 1 \ d\text{vol}_{n_0-k}$$

$$= \int_{S_{z_1,\ldots,z_k} \cap K} \mathbb{1}_{\mathcal{O}} \ d\text{vol}_{n_0-k}$$

$$= \int_{S_{z_1,\ldots,z_k} \cap K} \mathbb{1}_{\forall j: \ z_j \ \text{good at} \ x} \ d\text{vol}_{n_0-k}.$$

First equality is clear, in the second equality we use that:

$$\tilde{S}_{z_1,\ldots,z_k} \cap K = \left( \bigcap_{j=1}^{k} \tilde{S}_{z_j} \right) \cap K = \left( \bigcap_{j=1}^{k} S_{z_j} \cap \mathcal{O} \right) \cap K = \left( \bigcap_{j=1}^{k} S_{z_j} \cap K \right) \cap \mathcal{O}.$$

For the third equality, note that for all $x \in S_{z_1,\ldots,z_k} \cap K$, $x \in \mathcal{O}$ implies that there is a path of open neurons that connects from x to the output layer of the neural network. We also have that at $x$ all of the neurons $z_i \ i \in [k]$ satisfy $z_i(x) - b_i = 0$ So then we just need that there is a path from all of these neurons to the output later. Now we may re-write:

$$\mathbf{z} = \begin{pmatrix} z_1 \\ \vdots \\ z_k \end{pmatrix} \quad \mathbf{b} = \begin{pmatrix} b_1 \\ \vdots \\ b_k \end{pmatrix} \implies S_{z_1,\cdots,z_k} = \{x \in \mathbb{R}^n \ : \ \mathbf{z}(x) - \mathbf{b} = 0\}.$$

Then,

$$\text{vol}_{n_0-k} \left( \tilde{S}_{z_1,\ldots,z_k} \cap K \right) = \int_{\{\mathbf{z}(x)=\mathbf{b}\} \cap K} \mathbb{1}_{\forall j: \ z_j \ \text{good at} \ x} \ d\text{vol}_{n_0-k}.$$

For notational convenience let $\mathbf{b} = \beta_i + \delta_i$ Now recall that we have the Gaussian density function $\rho_{\mathbf{b}} : \mathbb{R}^k \to [0, 1]$ over the biases. Then we will first take expectations over $\mathbf{b}$, conditioned on the rest of the biases, which we will denote by $\hat{b}$:

$$\underset{\mathbf{b} \sim \rho_{\mathbf{b}}}{\mathbb{E}} \left[ \mathrm{vol}_{n_0 - k} \left( \tilde{S}_{z_1, \dots, z_k} \cap K \right) | \hat{b} \right] \tag{23}$$

$$= \int_{\mathbb{R}^k} \rho_{\mathbf{b}}(\mathbf{b}) \int_{\{\mathbf{z}(x) = \mathbf{b}\} \cap K} \mathbb{1}_{\forall j: \, z_j \text{ good at } x} \; d\mathrm{vol}_{n_0 - k}(x) \; d\mathbf{b} \tag{24}$$

$$= \int_{\mathbb{R}^k} \int_{\{\mathbf{z}(x) = \mathbf{b}\} \cap K} \rho_{\mathbf{b}}(\mathbf{z}(x)) \, \mathbb{1}_{\forall j: \, z_j \text{ good at } x} \; d\mathrm{vol}_{n_0 - k}(x) \; d\mathbf{b}. \tag{25}$$

To apply the co-area formula here, we take, borrowing notation from Appendix A.3.1, that:

$$u^{-1}(\mathbf{b}) = \{z(x) = \mathbf{b}\} \cap K.$$

So then,

$$u = z|_K,$$

and

$$g(x) = \rho_{\mathbf{b}}(\mathbf{z}(x)) \, \mathbb{1}_{\forall j: \, z_j \text{ good at } x}.$$

Notice $u$ is Lipschitz in $K$ and $g$ is dominated by an $L^1$ function $\rho_{\mathbf{b}}$ so we have that we may apply the co-area formula and we get:

$$\int_{\mathbb{R}^k} \int_{\{\mathbf{z}(x) = \mathbf{b}\} \cap K} \rho_{\mathbf{b}}(\mathbf{z}(x)) \, \mathbb{1}_{\forall j: \, z_j \text{ good at } x} \; d\mathrm{vol}_{n_0 - k}(x) \; d\mathbf{b}$$

$$= \int_K \| J_{\mathbf{z}}(x) \| \, \rho_{\mathbf{b}}(\mathbf{z}(x)) \, \mathbb{1}_{\forall j: \, z_j \text{ good at } x} \; dx.$$

We can now take expectations with respect to the remaining biases, since by the law of total expectation:

$$\underset{\tilde{\theta}}{\mathbb{E}} \, \underset{\mathbf{b} \sim \rho_{\mathbf{b}}}{\mathbb{E}} \left[ \mathrm{vol}_{n_0 - k} \left( \tilde{S}_{z_1, \dots, z_k} \cap K \right) | \hat{b} \right] = \underset{\tilde{\theta}}{\mathbb{E}} [\mathrm{vol}_{n_0 - k} \left( \tilde{S}_{z_1, \dots, z_k} \cap K \right)].$$

$\square$

### A.3.4 PROOF OF THEOREM 2

For the sake of readability, we restate the theorem here,

**Theorem 2.** *Let $\rho_{b_i}(x) = N(\beta_i, \sigma)$ be the density for the bias of neuron $z_i$. Then the following holds:*

$$\mathbf{LC} = \sum_{\text{neuron } z_i} \underset{x, \, \tilde{\theta}}{\mathbb{E}} \left[ \| \nabla z_i(x) \|_2 \, \rho_{b_i}(z_i(x)) \, \mathbb{1}_{z_i \text{ is good at } x} \right], \tag{6}$$

*where for each neuron the expectation is taken over $\tilde{\theta}$ and $x \sim p$.*

*Proof.* Recall first the definition of the local complexity density function $f$:

$$f(x) = \lim_{\epsilon \to 0} \frac{1}{Z_\epsilon} \underset{\tilde{\theta}}{\mathbb{E}} \left[ \mathrm{vol}_{n_0 - 1}(\mathcal{B}_{\mathcal{N}_{\tilde{\theta}}} \cap B_\epsilon(x)) \right]. \tag{26}$$

Now from here, we can compute, by using Lemma 11 in the second equality and using Lemma 12 fifth equality:

$$
\begin{aligned}
f(x) &= \lim_{\epsilon \to 0} \frac{1}{Z_\epsilon} \mathbb{E}_{\tilde{\theta}} \left[ \mathrm{vol}_{n_0-1}(B_\mathcal{N} \cap B_\epsilon(x)) \right] \\
&= \lim_{\epsilon \to 0} \frac{1}{Z_\epsilon} \mathbb{E}_{\tilde{\theta}} \left[ \mathrm{vol}_{n_0-1} \left( \bigcup_{z_i} \left( \tilde{S}_{z_i} \cap B_\epsilon(x) \right) \right) \right] \\
&= \lim_{\epsilon \to 0} \frac{1}{Z_\epsilon} \mathbb{E}_{\tilde{\theta}} \left[ \sum_{\text{neuron } z_i} \mathrm{vol}_{n_0-1} \left( \tilde{S}_{z_i} \cap B_\epsilon(x) \right) \right] \\
&= \sum_{\text{neuron } z_i} \lim_{\epsilon \to 0} \frac{1}{Z_\epsilon} \mathbb{E}_{\tilde{\theta}} \left[ \mathrm{vol}_{n_0-1} \left( \tilde{S}_{z_i} \cap B_\epsilon(x) \right) \right] \\
&= \sum_{\text{neuron } z_i} \lim_{\epsilon \to 0} \frac{1}{Z_\epsilon} \left( \int_{B_\epsilon(x)} \mathbb{E}_{\tilde{\theta}}[\|\nabla z_i(x)\| \, \rho_{b_i}(z_i(x)) \, \mathbb{1}_{z_i \text{ good at } x}] dx \right) \\
&= \sum_{\text{neuron } z_i} \mathbb{E}_{\tilde{\theta}}[\|\nabla z_i(x)\| \, \rho_{b_i}(z_i(x)) \, \mathbb{1}_{z_i \text{ good at } x}].
\end{aligned}
$$

In the last equality we use that the term $\mathbb{E}_{\tilde{\theta}} \left[ \|\nabla z_i(x)\| \, \rho_{b_i}(z_i(x)) \, \mathbb{1}_{z_i \text{ good at } x} \right]$ is continuous in $x$, which is a consequence of taking expectation over the biases. Taking expectation over $x \sim p$ completes the proof. $\qquad\square$

### A.4 PROOF OF COROLLARY 3

**Corollary 13.** *In the same setting as Theorem 2, let $C_{grad}$ be an upper bound on the norm of the gradient of every neuron $z_i$, $\|\nabla z_i(x)\| \leq C_{grad}$ for all $x \in \Omega$, $\tilde{\theta} = (W_1, \beta + \delta_1, \dots, W_L, \beta + \delta_L)$, let $C_{bias} = \frac{1}{\sqrt{2\pi}\sigma}$, and let $B = \mathbb{E}_{\tilde{\theta}, x \sim p} \left[ \sum_{\text{neuron } z_i} \mathbb{1}_{z_i \text{ not good at } x} \right]$ denote the expected number neurons that are not good. Then we have that:*

$$
\mathbf{LC} \leq C_{bias} \sum_{\text{neuron } z_i} \mathbb{E}_{\tilde{\theta}; x \sim p} \left[ \|\nabla z_i(x)\|_2 \right]. \tag{7}
$$

*Furthermore, for any $\eta > 0$ there are constants $c_{bias}^\eta = \frac{1}{\sqrt{2\pi}\sigma} e^{\frac{-\eta^2}{2\sigma^2}}$ and $\bar{\xi}_\eta = \Theta \left( e^{\frac{-\eta^2}{2\sigma^2}} / \eta^2 \right)^3$ such that:*

$$
\mathbf{LC} \geq c_{bias}^\eta \sum_{\text{neuron } z_i} \mathbb{E}_{\tilde{\theta}; x \sim p} \left[ \|\nabla z_i(x)\|_2 \right] - \bar{\xi}_\eta - B \cdot C_{grad} \cdot C_{bias}. \tag{8}
$$

*Proof.* For the upper bound, it is clear that we can write, assuming the conclusion of the prior theorem:

$$
\begin{aligned}
\mathbf{LC} &= \sum_{\text{neuron } z_i} \mathbb{E}_{\tilde{\theta}; x \sim p} \left[ \|\nabla z_i(x)\|_2 \, \rho_{b_{z_i}}(z_i(x)) \, \mathbb{1}_{z_i \text{ is good at } x} \right] \\
&\leq \sum_{\text{neuron } z_i} \mathbb{E}_{\tilde{\theta}; x \sim p} \left[ \|\nabla z_i(x)\|_2 \, \rho_{b_{z_i}}(z_i(x)) \right] \\
&\leq C_{bias} \sum_{\text{neuron } z_i} \mathbb{E}_{\tilde{\theta}; x \sim p} \left[ \|\nabla z_i(x)\|_2 \right].
\end{aligned}
$$

We can take $C_{grad} = \max_{\ell \in [L]} \|W_\ell W_{\ell-1} \cdots W_1\|_{op}$, which is clearly deterministic as it does not depend on the biases. To show the lower bound, we have the following bounds. Assuming the $C_{grad} \geq \|\nabla z_i(x)\|^2$ for all neurons $z_i$, $x \in \Omega$ and $C_{bias} \geq \rho_b$ and that on average $B$ neurons are not good at $x \sim p$:

$$
B = \mathbb{E}_{x \sim p} \mathbb{E}_{\tilde{\theta}} \left[ \sum_{z_i \text{ neuron}} \mathbb{1}_{z_i \text{ not good at } x} \right].
$$

---

[3]We use the standard Big Theta notation $f(x) = \Theta(g(x))$ to signify that there exist $c_1, c_2, x_0$ such that $c_1 g(x) \leq f(x) \leq c_2 g(x)$ for all $x > x_0$.

It is clear then that we can bound the local complexity as:

$$\mathbf{LC} = \mathbb{E}_{x \sim p, \tilde{\theta}} \left[ \sum_{\text{neuron } z_i} \|\nabla z_i(x)\| \, \rho_{b_{z_i}}(z_i(x)) - \sum_{\text{neuron } z_i \text{ not good at } x} \|\nabla z_i(x)\| \, \rho_{b_{z_i}}(z_i(x)) \right] \quad (27)$$

$$\geq \sum_{\text{neuron } z_i} \mathbb{E}_{x \sim \tilde{\theta}} \left[ \|\nabla z_i(x)\| \, \rho_{b_{z_i}}(z_i(x)) \right] - B \cdot C_{\text{grad}} C_{\text{bias}} \quad (28)$$

$$\geq c_{\text{bias}}^{\eta} \sum_{\text{neuron } z_i} \mathbb{E}_{x \sim \tilde{\theta}} \left[ \|\nabla z_i(x)\| \right] - \bar{\xi}_{\eta} - B \cdot C_{\text{grad}} C_{\text{bias}}. \quad (29)$$

Where for the last inequality we proceed as follows: Take neuron $z$ with $\rho_b$ being the density for a Gaussian with variance $\sigma$ centered at $\beta$. Then:

$$\mathbb{E}_{x,\tilde{\theta}} \left[ \|\nabla z(x)\| \rho_b(z(x)) \right] \geq \mathbb{E}_{x,\tilde{\theta}} \left[ \|\nabla z(x)\| \, \rho_b(z(x)) \mathbb{1}_{|z(x)-b|\leq\eta} \right]$$

$$\geq \left[ \inf_{|r-b|\leq\eta} \{\rho_b(r)\} \right] \mathbb{E}_{x,\tilde{\theta}} \left[ \|\nabla z(x)\| \, \mathbb{1}_{|z(x)-b|\leq\eta} \right]$$

$$\geq \left[ \inf_{|r-b|\leq\eta} \{\rho_b(r)\} \right] \left( \mathbb{E}_{x,\tilde{\theta}} \left[ \|\nabla z(x)\| \right] - \mathbb{E}_{x,\tilde{\theta}} \left[ \|\nabla z(x)\| \, \mathbb{1}_{|z(x)-b|>\eta} \right] \right).$$

Notice we can bound the second term here as follows, using Markov's inequality:

$$\mathbb{E}_{x,\tilde{\theta}} \left[ \|\nabla z(x)\| \, \mathbb{1}_{|z(x)-b|>\eta} \right] \leq C_{\text{grad}} \mathbb{P}_{x,\tilde{\theta}}(|z(x) - b| \geq \eta)$$

$$\leq C_{\text{grad}} \frac{\mathbb{E}_{x,\tilde{\theta}}[|z(x) - b|^2]}{\eta^2}$$

$$\leq C_{\text{grad}} \frac{\mathbb{E}_{x,\tilde{\theta}}[z(x)^2] + \mathbb{E}_{x,\tilde{\theta}}[b^2]}{\eta^2}.$$

Now since the data distribution has compact support, we have that $\mathbb{E}_{x,\tilde{\theta}}[z(x)^2]$ and $\mathbb{E}_{x,\tilde{\theta}}[b^2]$ are uniformly bounded. This gives us that,

$$\mathbb{E}_{x,\tilde{\theta}} \left[ \|\nabla z(x)\| \rho_b(z(x)) \right] \geq c_{\text{bias}}^{\eta} \left( \mathbb{E}_{x,\tilde{\theta}} \left[ \|\nabla z(x)\| \right] - \xi(\eta, \sigma, z) \right),$$

where $\xi(\eta, \sigma, z) = \Theta(\frac{1}{\eta^2})$, $c_{\text{bias}}^{\eta} = \frac{1}{\sqrt{2\pi}\sigma} e^{\frac{-\eta^2}{2\sigma^2}}$. Now define

$$\bar{\xi}(\eta, \sigma, \mathcal{N}) = c_{\text{bias}}^{\eta} \sum_{z \text{ neuron}} \xi(\eta, \sigma, z) = \Theta \left( \frac{e^{\frac{-\eta^2}{2\sigma^2}}}{\eta^2} \right).$$

Taking a sum over every neuron then gives that

$$\mathbb{E}_{x \sim p, \delta} \left[ \sum_{\text{neuron } z_i} \left[ \|\nabla z_i(x)\| \, \rho_{b_{z_i}}(z_i(x)) \right] \right] \geq c_{\text{bias}}^{\eta} \mathbb{E}_{x \sim p, \delta} \left[ \sum_{\text{neuron } z_i} |\nabla z_i(x)\| \right] - \bar{\xi}(\eta, \sigma, \mathcal{N}),$$

where $\bar{\xi}(\eta, \sigma, \mathcal{N}) = \Theta(\frac{e^{\frac{-\eta^2}{2\sigma}}}{\eta^2})$. We abbreviate this as $\bar{\xi}_{\eta}$ in later results. Using this result in (29) completes the proof. $\qquad \square$

## A.5 PROOF OF THEOREM 5

We first recall from before that we define $\text{rank}_{\epsilon}(\text{Jac}(\mathbf{z}_l))$ to be the number of singular values of $\text{Jac}(\mathbf{z}_l)$ bigger than $\epsilon$. We define the approximate local rank to be:

$$\mathbf{LR}_l^{\epsilon} = \mathbb{E}_{x \sim p} \left[ \text{rank}_{\epsilon}(J_x \mathbf{z}_l(x)) \right].$$

**Theorem 5.** *For any $\epsilon > 0$, the local ranks across layers can be bounded in terms of the local complexity as follows:*

$$\frac{1}{n_0 \, C_{bias}} \, \mathbf{LC} \le \sum_{l=1}^{L} \sqrt{C_{grad}^2 \, \mathbf{LR}_l^\epsilon + \epsilon^2 n_l}. \tag{11}$$

*Moreover, in the same setting as in Corollary 3,*

$$\sum_{l=1}^{L} \mathbf{LR}_l^\epsilon \le \frac{1}{c_{bias}^\eta \epsilon^2} \left[ \mathbf{LC} + \bar{\xi}_\eta + B \cdot C_{grad} \cdot C_{bias} \right]. \tag{12}$$

*Proof.* Notice that we have immediately, for an $n$ by $n$ matrix $A$ with $\text{rank}_\epsilon = m$,

$$\epsilon^2 m \le \sum_{i=0}^{n} \sigma_i(A)^2 \le \|A\|_F^2 = \sum_{i=0}^{n} \sigma_i(A)^2 \le m \sigma_{\max}(A)^2 + (n-m)\epsilon \le m \sigma_{\max}(A)^2 + n\epsilon. \tag{30}$$

Notice also that we have that, using that $\sqrt{a + b} \le \sqrt{a} + \sqrt{b}$:

$$\|J\mathbf{z}_l(x)\|_F = \sqrt{\sum_{\text{neuron } z_i \in \text{layer } l} \|\nabla z_i(x)\|_2^2} \le \sum_{\text{neuron } z_i \in \text{layer } l} \|\nabla z_i(x)\|.$$

So we may write that:

$$\text{rank}_\epsilon(J\mathbf{z}_l(x)) \le \frac{1}{\epsilon^2} \sum_{\text{neuron } z_i \in \text{layer } l} \|\nabla z_i(x)\|_2.$$

Summing this over all layers $l \in [L]$ and taking expectation over the data distribution and $\tilde{\theta}$ gives us:

$$\sum_{l=1}^{L} \mathbf{LR}_l^\epsilon \le \frac{1}{\epsilon^2} \mathop{\mathbb{E}}_{\tilde{\theta}, x \sim p} \left[ \sum_{\text{neuron } z_i} \|\nabla z_i(x)\|_2 \right].$$

Now recall that from Corollary 3 we have that:

$$\mathbf{LC} \ge c_{\text{bias}}^\eta \sum_{\text{neuron } z_i} \mathop{\mathbb{E}}_{\tilde{\theta}; x \sim p} \left[ \|\nabla z_i(x)\|_2 \right] - \bar{\xi}_\eta - B \cdot C_{\text{grad}} \cdot C_{\text{bias}}.$$

Which is equivalent to:

$$\mathop{\mathbb{E}}_{\tilde{\theta}, x \sim p} \left[ \sum_{\text{neuron } z_i} \left[ \|\nabla z_i(x)\|_2 \right] \right] \le \frac{1}{c_{\text{bias}}^\eta} [\mathbf{LC} + \bar{\xi}_\eta + B \cdot C_{\text{grad}} C_{\text{bias}}].$$

Which gives us, as desired:

$$\sum_{l=1}^{L} \mathbf{LR}_l^\epsilon \le \frac{1}{\epsilon^2 c_{\text{bias}}^\eta} [\mathbf{LC} + \bar{\xi}_\eta + B \cdot C_{\text{grad}} C_{\text{bias}}].$$

Recall that $C_{\text{grad}} = \max_{\ell \in [L]} \|W_\ell W_{\ell-1} \cdots W_1\|_{op}$. Now, for the other inequality we need first the following two sub-claims:

**Claim 1:**

$$\mathop{\mathbb{E}}_{x \sim p; \tilde{\theta}} \|J_x \mathbf{z}_l(x)\|_F \le \sqrt{C_{\text{grad}}^2 \mathbf{LR}_l^\epsilon + \epsilon^2 n_l}.$$

*Proof.*

$$\|J_x \mathbf{z}_l(x)\|_F^2 = \sum_{i=1}^{n_0} \sigma_i^2 \le \sigma_{\max}^2(J_x \mathbf{z}_l(x)) \, \text{rank}_\epsilon(J_x \mathbf{z}_l(x)) + \epsilon^2(n_l - \text{rank}_\epsilon(J_x \mathbf{z}_l(x)))$$

$$\le \sigma_{\max}^2(J_x \mathbf{z}_l(x)) \, \text{rank}_\epsilon(J_x \mathbf{z}_l(x)) + \epsilon^2 n_l$$

$$\le C_{\text{grad}}^2 \, \text{rank}_\epsilon(J_x \mathbf{z}_l(x)) + \epsilon^2 n_l.$$

Taking expectations with respect to $x \sim p$ and $\tilde{\theta}$ gives us $\mathbb{E}_{x \sim p; \tilde{\theta}} \|J_x \mathbf{z}_l(x)\|_F^2 \leq C_{\text{grad}} \mathbf{LR}_l^\epsilon + \epsilon^2 n_l$.
Now notice that Jenson's inequality gives us that $(\mathbb{E}_{x \sim p; \tilde{\theta}} \|J_x \mathbf{z}_l(x)\|_F)^2 \leq \mathbb{E}_{x \sim p; \tilde{\theta}} \|J_x \mathbf{z}_l(x)\|_F^2$,
which completes the proof after taking square roots on both sides. $\qquad\square$

**Claim 2:**
$$\mathbb{E}_{x \sim p; \tilde{\theta}} \|J_x \mathbf{z}_l(x)\|_F \geq \frac{1}{n_0} \sum_{z_i \text{neuron in layer } l} \mathbb{E}_{x \sim p; \tilde{\theta}} \|\nabla_x z_i(x)\|_2.$$

*Proof.*
$$\|J_x \mathbf{z}_l(x)\|_F \geq \|J_x \mathbf{z}_l(x)\|_2$$
$$\geq \frac{1}{\sqrt{n_0}} \|J_x \mathbf{z}_l(x)\|_\infty$$
$$= \frac{1}{\sqrt{n_0}} \sum_{i \in [n_l]} \|\nabla_x z_i(x)\|_1$$
$$= \frac{1}{\sqrt{n_0}} \sum_{i \in [n_l]} \frac{1}{\sqrt{n_0}} \|\nabla_x z_i(x)\|_2$$
$$= \frac{1}{n_0} \sum_{i \in [n_l]} \|\nabla_x z_i(x)\|_2.$$

This completes the proof of the subclaim after taking expectations on both sides. $\qquad\square$

Now we may prove our bound. Recall that the Local Complexity satisfies:
$$\mathbf{LC} \leq C_{\text{bias}} \sum_{\text{neuron } z_i} \mathbb{E}_{x \sim p; \tilde{\theta}} [\|\nabla z_i(x)\|].$$

So then we have that, using Claim (2),
$$\frac{1}{n_0 \, C_{\text{bias}}} \mathbf{LC} \leq \sum_{l \in [L]} \mathbb{E}_{x \sim p; \tilde{\theta}} \|J_x \mathbf{z}_l(x)\|_F.$$

And then by using Claim (1),
$$\frac{1}{n_0 \, C_{\text{bias}}} \mathbf{LC} \leq \sum_{l \in [L]} \sqrt{C_{\text{grad}}^2 \mathbf{LR}_l^\epsilon + \epsilon^2 n_l}.$$

Which concludes this proof. $\qquad\square$

### A.6 PROOF OF COROLLARY 14

Recall that $\mathbf{LR}_l = \mathbb{E}_{x \sim p} [\text{rank}(J_x \mathbf{z}_l(x))]$. Now we can restate the corollary:

**Corollary 14.** *In the same setting as Theorem 5:*
$$\frac{1}{n_0 \, C_{bias}} \mathbf{LC} \leq C_{grad} \sum_{l=1}^{L} \sqrt{\mathbf{LR}_l}. \tag{31}$$

*Proof.* The first inequality follows from the first inequality in Theorem 5, as well as by application of the fact that $\lim_{\epsilon \to 0} \mathbf{LR}_l^\epsilon = \mathbf{LR}_l$. Then notice that:
$$\frac{1}{n_0 \, C_{\text{bias}}} \mathbf{LC} \leq \sum_{l=1}^{L} \sqrt{C_{\text{grad}}^2 \mathbf{LR}_l^\epsilon + \epsilon^2 n_l} \xrightarrow[\epsilon \to 0]{} C_{\text{grad}} \sum_{l=1}^{L} \sqrt{\mathbf{LR}_l}.$$

$\qquad\square$

## A.7 Proof of Proposition 6

**Proposition 15.** *Suppose our data distribution admits a density function $p$ with support $\Omega$. Consider a point $\bar{x}$ in the interior of $\Omega$, with classification margin $\mathcal{N}_\theta(\bar{x}) > \gamma$. For any $\epsilon > 0$ with $B_\epsilon(\bar{x}) \in \Omega$, let $c_\epsilon = \inf_{x \in B_\epsilon(\bar{x})} p(x)$. Then $\mathbf{TV} \leq c_\epsilon \gamma$ implies there are no adversarial examples in $B_\epsilon(\bar{x})$.*

*Proof.* Let $\tilde{TV} = \int_{B_\epsilon(\bar{x})} |\mathcal{N}'_\theta(x)| dx$, and recall our original definition that,

$$TV = \int_\Omega \rho(x)|\mathcal{N}'_\theta(x)| dx.$$

Then we can clearly see that we have the following bound:

$$c_\epsilon \tilde{TV} \leq TV.$$

Now, via the contrapositive argument, suppose that we have $x \in B_\epsilon(\bar{x})$ an adversarial example, then,

$$\mathcal{N}_\theta(\bar{x}) - \mathcal{N}_\theta(x) > \gamma.$$

From here it is clear that $\tilde{TV} > \gamma$. So in particular,

$$TV > c_\epsilon \gamma.$$

This completes the proof. $\qquad\square$

## A.8 Proof of Theorem 7

**Theorem 7.** *Let $g_l$ denote the rest of the network after the $l$th layer, so that $\mathcal{N}_\theta = g_l \circ \phi(\mathbf{z}_l - b_l)$. Let $C_l$ denote the Lipschitz constant of $g_l$. Then with the same setting and notations as Theorem 2:*

$$\mathbf{TV} \cdot \frac{Lc_{bias}^\eta}{\max_{1 \leq l \leq L} C_l} - \bar{\xi}_\eta - B \cdot C_{grad} \cdot C_{bias} \leq \mathbf{LC}. \tag{13}$$

*Proof.* Following the notational conventions in Appendix A.1, recall for any layer $1 \leq l \leq L$, that our network is:

$$\mathcal{N}(x) = g_l \circ h_l(x). \tag{32}$$

Where $g_l$ denotes the rest of the network after layer $l$. Expanding a layer yields:

$$N(x) = g_l(\phi(W_l h_{l-1}(x) - b_l)). \tag{33}$$

Recall from Appendix A.1 that we write, where $n_l$ denotes the number of neurons at layer $l$.

$$W_l h_{l-1}(x) = \begin{pmatrix} \vdots \\ z_l^i(x) \\ \vdots \end{pmatrix}_{i \in [n_l]}.$$

Computing gradients on (33), we can get that:

$$\nabla_x \mathcal{N}(x) = \frac{\partial g_l}{\partial h_l} \frac{\partial h_l}{\partial x}$$

$$= \begin{pmatrix} \cdots & \nabla_x z_i^{(l)}(x) & \cdots \end{pmatrix}_{i \in [n_l]} \nabla_{h_l} g_l(h_l(x)) \odot \begin{pmatrix} \vdots \\ \mathbb{1}_{\{z_i^{(l)}(x) \geq b_l^i\}} \\ \vdots \end{pmatrix}_{i \in [n_l]}.$$

Now let $C_l$ denote the minimal Lipschitz constant for $g_l$ in the image of the data support $h_l(\Omega)$. Recall also the fact that $\|Av\|_2 \leq \|A\|_F \|v\|_2$. Now we can write that:

$$\underset{x \sim p}{\mathbb{E}} \left[ \|\nabla_x \mathcal{N}(x)\| \right]$$

$$= \underset{x \sim p}{\mathbb{E}} \left[ \left\| \overbrace{\left( \cdots \quad \nabla_x z_i^{(l)}(x) \quad \cdots \right)}^{A} \overbrace{\nabla_{h_l} g_l(h_l(x)) \odot \begin{pmatrix} \vdots \\ \mathbb{1}_{\{z_i^{(l)}(x) \geq b_i^i\}} \\ \vdots \end{pmatrix}_{i \in [n_l]}}^{v} \right\| \right]$$

$$\leq \underset{x \sim p}{\mathbb{E}} \left[ \left\| \nabla_{h_l} g_l(h_l(x)) \odot \begin{pmatrix} \vdots \\ \mathbb{1}_{\{z_i^{(l)}(x) \geq b_i^i\}} \\ \vdots \end{pmatrix}_{i \in [n_l]} \right\| \left( \sum_{i \in [n_l]} \|\nabla_x z_i^{(l)}(x)\|^2 \right)^{\frac{1}{2}} \right]$$

$$\leq \underset{x \sim p}{\mathbb{E}} \left[ \|\nabla_{h_l} g_l(h_l(x))\| \left( \sum_{i \in [n_l]} \|\nabla_x z_i^{(l)}(x)\|^2 \right)^{\frac{1}{2}} \right]$$

$$\leq C_l \underset{x \sim p}{\mathbb{E}} \left[ \left( \sum_{i \in [n_l]} \|\nabla_x z_i^{(l)}(x)\|^2 \right)^{\frac{1}{2}} \right]$$

$$\leq C_l \underset{x \sim p}{\mathbb{E}} \left[ \sum_{i \in [n_l]} \|\nabla_x z_i^{(l)}\| \right].$$

Where in the last inequality we use that $\sqrt{a + b} \leq \sqrt{a} + \sqrt{b}$. Now applying this inequality to each of the $L$ total layers and taking the sum, we get that:

$$L \underset{x \sim p}{\mathbb{E}} \left[ \|\nabla_x \mathcal{N}(x)\| \right] \leq \sum_{l=1}^{L} C_l \underset{x \sim p}{\mathbb{E}} \left[ \sum_{i \in [n_l]} \|\nabla_x z_i^{(l)}(x)\| \right]$$

$$\leq \max_{1 \leq l \leq L} C_l \underset{x \sim p}{\mathbb{E}} \left[ \sum_{z_i \text{ neuron}} \|\nabla_x z_i(x)\| \right].$$

Now take expectations over $\tilde{\theta}$, and we can combine this with the bound from before in Corollary 3,

$$\underset{x \sim p, \tilde{\theta}}{\mathbb{E}} \left[ \sum_{\text{neuron } z_i} [\|\nabla z_i(x)\|_2] \right] \leq \frac{1}{c_{\text{bias}}} [\mathbf{LC} + \bar{\xi}_\eta + B \cdot C_{\text{grad}} C_{\text{bias}}].$$

Which then gives us:

$$\frac{L}{\max_{1 \leq l \leq L} C_l} \underset{x \sim p, \tilde{\theta}}{\mathbb{E}} \left[ \|\nabla_x \mathcal{N}(x)\| \right] \leq \frac{1}{c_{\text{bias}}} [\mathbf{LC} + \bar{\xi}_\eta + B \cdot C_{\text{grad}} C_{\text{bias}}],$$

as desired. $\qquad \square$

## A.9 PROOF OF PROPOSITION 8

The representation cost is defined as $R_\Omega(f) = \inf_{\theta: \, \mathcal{N}_\theta(\Omega) = f(\Omega)} \|\theta\|_F$. We re-state our main proposition:

**Proposition 16.** *In the same setting as Theorem 2, where $n_l$ is the maximum hidden layer dimension,*

$$\frac{n_0}{C_{bias}} \mathbf{LC} \leq n_l^{\frac{1-L}{2}} L^{1 - \frac{L}{2}} R(\mathcal{N}_\theta)^L. \tag{14}$$

*Proof.* We begin by computing that:

$$J\mathbf{z}_l(x) = W_l D_l W_{l-1} D_{l-1} \cdots D_1 W_1.$$

With $W_l$ denoting the $l$-th layer weight matrix and $D_l$ being a diagonal matrix of 0 and 1 denoting the ReLU activation pattern at the $l-$th layer, when evaluated at $x$. Now using a result from Soudry et al. (2018) and Jacot (2023a) we can get that, for $p = \frac{2}{L}$ ($\|\cdot\|_p$ denotes the $L_p$ Schatten matrix norm):

$$\|J\mathbf{z}_l(x)\|_p^p \leq \frac{1}{L}\left(\|W_l D_l\|_F^2 + \|W_{l-1} D_{l-1}\|_F^2 \cdots \|D_1 W_1\|\right) \tag{34}$$

$$\leq \frac{1}{L}\left(\|W_l\|_F^2 + \|W_{l-1}\|_F^2 \cdots \|W_1\|\right) \tag{35}$$

$$\leq \frac{1}{L}\|\theta\|_F^2. \tag{36}$$

Now we recall the equivalence of the $L_p$ Schatten matrix norm to the Frobenius norm. Notice this is the same as the equivalence suffices to do this for the equivalent vectors of singular values for the respective norms. For any $n \times n$ matricies $A, B$:

$$\|A\|_F \leq C\|B\|_p.$$

Where $C = n^{\frac{1}{2} - \frac{1}{p}} = n^{\frac{1-L}{2}}$. Then we also have that:

$$n^{\frac{L-1}{2}}\|A\|_F \leq \|B\|_p \implies (n^{\frac{L-1}{2}}\|A\|_F)^{\frac{2}{L}} \leq \|B\|_p^p.$$

Applying this to (36) gives us:

$$(n_l^{\frac{L-1}{2}}\|J\mathbf{z}_l(x)\|_F)^{\frac{2}{L}} \leq \frac{1}{L}\|\theta\|_F^2 \implies \|J\mathbf{z}_l(x)\|_F \leq n_l^{\frac{1-L}{2}}(\frac{1}{L})^{\frac{L}{2}}\|\theta\|_F^L.$$

Since this holds for all parameterizations of the function learned by $\mathcal{N}$, we can get that:

$$\|J\mathbf{z}_l(x)\|_F \leq n_l^{\frac{1-L}{2}} L^{\frac{-L}{2}} R(\mathcal{N})^L.$$

Apply this, summing over all layers to get:

$$\sum_{l=1}^{L} \|J\mathbf{z}_l(x)\|_F \leq n_l^{\frac{1-L}{2}} L^{1-\frac{L}{2}} R(\mathcal{N})^L.$$

Now combine this bound with (A.5) and we get:

$$\frac{n_0}{C_{\text{bias}}}\mathbf{LC} \leq n_l^{\frac{1-L}{2}} L^{1-\frac{L}{2}} R(\mathcal{N})^L.$$

$\square$

## A.10   PROOF OF PROPOSITION 17

Canonical results on training neural networks in the lazy/kernel regime show that the weights do not move far from their initialization by the end of training (Chizat et al., 2019). The following proposition shows that if this holds, then the local complexity will also not change much from the beginning to the end of training.

**Proposition 17.** *Consider a 2-layer MLP of the form $\mathcal{N}_\theta(x) = v^T \phi(Wx - \beta)$ with parameters $\theta_0 = (\beta^{(0)}, W^{(0)}, v^{(0)})$ at initialization and $\theta_t = (\beta^{(t)}, W^{(t)}, v^{(t)})$ at time $t$. Suppose also that $\|\theta_0 - \theta_t\|_2 \leq \epsilon$. Denote by $\mathbf{LC}(\theta_t)$ the local complexity of parameters $\theta_t$. If $v \neq 0$, then, we have that there exists a constant $C$ independent of $\epsilon$ such that,*

$$|\mathbf{LC}(\theta_t) - \mathbf{LC}(\theta_0)| \leq C\epsilon$$

*Proof.* First note that in the setting of this 2-layer network we can apply Theorem 2 and see that the local complexity is,

$$\mathbf{LC}(\theta_0) = \frac{1}{\sqrt{2\pi}\sigma} \sum_{k=1}^{n_1} \mathbb{E}_{x \sim p} \left[ \|w_k^{(0)}\| e^{-\frac{(\langle w_k^{(0)}, x \rangle - \beta_k^{(0)})^2}{2\sigma^2}} \right].$$

Where we denote that $w_k$ is the $k$-th row of $W$. Then notice that,

$$\left| \frac{1}{\sqrt{2\pi}\sigma} e^{-\frac{(\langle w_k^{(0)}, x \rangle - \beta_k^{(0)})^2}{2\sigma^2}} \right| \leq \frac{1}{\sqrt{2\pi}\sigma}.$$

So then, we can compute that

$$|\mathbf{LC}(\theta_0) - \mathbf{LC}(\theta_t)|$$

$$= |\frac{1}{\sqrt{2\pi}\sigma} \sum_{k=1}^{n_1} \mathbb{E}_{x \sim p} \left[ \|w_k^{(0)}\| e^{-\frac{(\langle w_k^{(0)}, x \rangle - \beta_k^{(0)})^2}{2\sigma^2}} \right] - \frac{1}{\sqrt{2\pi}\sigma} \sum_{k=1}^{n_1} \mathbb{E}_{x \sim p} \left[ \|w_k^{(t)}\| e^{-\frac{(\langle w_k^{(t)}, x \rangle - \beta_k^{(t)})^2}{2\sigma^2}} \right] |$$

$$\leq \frac{1}{\sqrt{2\pi}\sigma} |\sum_{k=1}^{n_1} \|w_k^{(0)}\| - \|w_k^{(t)}\||$$

$$\leq \frac{1}{\sqrt{2\pi}\sigma} \sum_{k=1}^{n_1} \epsilon$$

$$\leq \frac{n_1}{\sqrt{2\pi}\sigma} \epsilon.$$

$\square$

## A.11 PROOF OF PROPOSITION 9

We first recall a theorem courtesy of Timor et al. (2023):

**Theorem 18.** *[Timor et al., 2023] Let $\{(x_i, y_i)\}_{i=1}^n \subseteq \mathbb{R}^{d_{in}} \times \{-1, 1\}$ be a binary classification dataset, and assume that there is $i \in [n]$ with $\|x_i\| \leq 1$. Assume that there is a fully-connected neural network $N$ of width $m \geq 2$ and depth $k \geq 2$, such that for all $i \in [n]$ we have $y_i N(x_i) \geq 1$, and the weight matrices $W_1, \ldots, W_k$ of $N$ satisfy $\|W_i\|_F \leq B$ for some $B > 0$. Let $N_\theta$ be a fully-connected neural network of width $m' \geq m$ and depth $k' > k$ parameterized by $\theta$. Let $\theta^* = [W_1^*, \ldots, W_L^*]$ be a global optimum of the above optimization problem (15). Namely, $\theta^*$ parameterizes a minimum-norm fully-connected network of width $n_l$ and depth $L$ that labels the dataset correctly with margin 1. Then, we have*

$$\frac{1}{L} \sum_{i=1}^L \frac{\|W_i^*\|_{op}}{\|W_i^*\|_F} \geq \frac{1}{\sqrt{2}} \cdot \left( \frac{\sqrt{2}}{B} \right)^{\frac{k}{L}} \cdot \sqrt{\frac{L}{L+1}}. \tag{37}$$

*Equivalently, we have the following upper bound on the harmonic mean of the ratios $\frac{\|W_i^*\|_F}{\|W_i^*\|_{op}}$:*

$$\frac{L}{\sum_{i=1}^L \left( \frac{\|W_i^*\|_F}{\|W_i^*\|_{op}} \right)^{-1}} \leq \sqrt{2} \cdot \left( \frac{B}{\sqrt{2}} \right)^{\frac{k}{L}} \cdot \sqrt{\frac{L+1}{L}}. \tag{38}$$

We will leverage the result in this theorem, particularly the bound on the harmonic mean of the ratios $\frac{\|W_i^*\|_F}{\|W_i^*\|_{op}}$ to prove the following proposition.

**Proposition 19.** *Let $\{(x_i, y_i)\}_{i=1}^n \subseteq \mathbb{R}^{n_0} \times \{-1, 1\}$ be a binary classification dataset, and assume that there is $i \in [n]$ with $\|x_i\| \leq 1$. Assume that there is a fully-connected neural network $\mathcal{N}$ of width $m \geq 2$ and depth $k \geq 2$, such that for all $i \in [n]$ we have $y_i \mathcal{N}(x_i) \geq 1$, and the weight matrices $W_1, \ldots, W_k$ of $\mathcal{N}$ satisfy $\|W_i\|_F \leq B$ for some $B > 0$. Let $\mathcal{N}_\theta$ be a fully-connected neural network of width $m' \geq m$ and depth $k' > k$ parameterized by $\theta$. Let $\theta^* = [W_1^*, \ldots, W_L^*]$*

*be a global optimum of the above optimization problem (15). Then, assuming the same setting as Theorem 2, we have the following bound on the local complexity:*

$$\frac{1}{L \max_{l \in [L]} \|W_i^*\|_{op}} \left( \frac{n_0}{C_{bias} \, n_l^{\frac{1-L}{2}} L^{1-\frac{L}{2}}} \mathbf{LC} \right)^{\frac{1}{L}} - \gamma \leq \sqrt{2} \cdot \left( \frac{B}{\sqrt{2}} \right)^{\frac{k}{L}} \cdot \sqrt{\frac{L+1}{L}}, \qquad (16)$$

*where,* $\gamma = \|W_i^*\|_F \left( \sqrt{\frac{1}{\|W_l^*\|_{op}}} - \sqrt{\frac{1}{\|W_i^*\|_{op}}} \right)^2.$

*Proof.* Following an intermediate result from Section A.9 gives us that, for $K = n_l^{\frac{1-L}{2}} L^{1-\frac{L}{2}}$:

$$\frac{n_0}{C_{\text{bias}} \, K} \mathbf{LC} \leq \left( \sum_{i \in [k']} \|W_i^*\|_F \right)^L \implies \left( \frac{n_0}{C_{\text{bias}} \, K} \mathbf{LC} \right)^{\frac{1}{L}} \leq \sum_{i \in [k']} \|W_i^*\|_F.$$

Then we can see that we also would have:

$$\frac{1}{L \max_{l \in [L]} \|W_i^*\|_{\text{op}}} \left( \frac{n_0}{C_{\text{bias}} \, K} \mathbf{LC} \right)^{\frac{1}{L}} \leq \frac{1}{L} \sum_{i \in [L]} \frac{\|W_i^*\|_F}{\|W_i^*\|_{\text{op}}}.$$

Now via the bound controlling the difference between the arithmetic mean and the harmonic mean from Meyer (1984), we can get that:

$$\frac{1}{L} \sum_{i \in [L]} \frac{\|W_i^*\|_F}{\|W_i^*\|_{\text{op}}} - \frac{L}{\sum_{i=1}^{L} \left( \frac{\|W_i^*\|_F}{\|W_i^*\|_{\text{op}}} \right)^{-1}} \leq (\sqrt{\alpha_{\max}} - \sqrt{\alpha_{\min}})^2.$$

Where,

$$\alpha_{\max} = \max_{l \in [k']} \frac{\|W_l^*\|_F}{\|W_l^*\|_{\text{op}}},$$

and

$$\alpha_{\min} = \min_{i \in [k']} \frac{\|W_i^*\|_F}{\|W_i^*\|_{\text{op}}}.$$

But notice that, by Lemma 15 in Timor et al. (2023), we have that:

$$\|W_l^*\|_F = \|W_i^*\|_F.$$

So then,

$$(\sqrt{\alpha_{\max}} - \sqrt{\alpha_{\min}})^2 = \|W_i^*\|_F \left( \sqrt{\frac{1}{\|W_l^*\|_{\text{op}}}} - \sqrt{\frac{1}{\|W_i^*\|_{\text{op}}}} \right)^2 = \gamma.$$

and we get as a consequence of the bound controlling the Harmonic Mean from the prior theorem:

$$\frac{1}{L \max_{l \in [L]} \|W_i^*\|_{\text{op}}} \left( \frac{n_0}{C_{\text{bias}} \, K} \mathbf{LC} \right)^{\frac{1}{L}} - \gamma \leq \sqrt{2} \cdot \left( \frac{B}{\sqrt{2}} \right)^{\frac{k}{L}} \cdot \sqrt{\frac{L+1}{L}}.$$

$\square$

## A.12  DERIVATION OF COROLLARY 10 (INFORMAL)

**Corollary 20.** *[Informal] Suppose that* $\|\theta(t)\|_2 = \Theta \left( (\log \frac{1}{\lambda})^{1/L} \right)$ *holds. Then, in the "rich" phase of training the local complexity is bounded:*

$$\frac{n_0}{C_{bias} n_l^{\frac{1-L}{2}} L^{1-\frac{L}{2}}} \mathbf{LC} \leq \Theta(\log \frac{1}{\lambda}). \qquad (17)$$

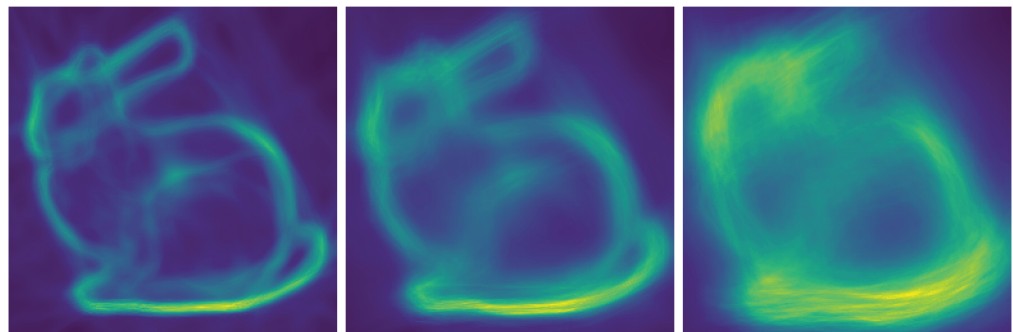

Figure 4: Here we show the effects of estimating the local complexity density function $f$ with varying levels of $\sigma$. We show $\sigma = 0.025$ (Left), $\sigma = 0.05$ (Middle), and $\sigma = 0.1$ (Right).

Suppose that $\|\theta(t)\|_2 = \Theta((\log \frac{1}{\lambda})^{1/L})$. Then recall by Proposition 6 we have that:

$$\frac{n_0}{C_{\text{bias}}} \mathbf{LC} \leq n_l^{\frac{1-L}{2}} L^{1-\frac{L}{2}} R(\mathcal{N})^L$$

$$\leq n_l^{\frac{1-L}{2}} L^{1-\frac{L}{2}} \|\theta(t)\|_2^L$$

$$= n_l^{\frac{1-L}{2}} L^{1-\frac{L}{2}} \Theta((\log \frac{1}{\lambda})^{1/L})^L$$

$$= n_l^{\frac{1-L}{2}} L^{1-\frac{L}{2}} \Theta(\log \frac{1}{\lambda}).$$

## B  MORE INFORMATION ON EMPIRICAL STUDIES

### B.1  ON ESTIMATION OF THE LOCAL COMPLEXITY IN FIGURE 1

The network in question is trained to exactly represent a 2D grayscale image of the Stanford Bunny Turk & Levoy (1994), using the Mean Squared Error loss function and Adam optimizer with learning rate $1e - 4$. The left hand figure is an exact visualizations of the linear regions in this network computed using Humayun et al. (2023a).

To understand how we compute the local complexity, let us first recall the key result from Theorem 2, from which we then use the trivial upper bound on the indicator function:

$$\mathbf{LC} = \sum_{\text{neuron } z_i} \mathbb{E}_{x,\tilde{\theta}} [\|\nabla z_i(x)\|_2 \, \rho_{b_i}(z_i(x)) \, \mathbb{1}_{z_i \text{ is good at } x}] \leq \sum_{\text{neuron } z_i} \mathbb{E}_{x,\tilde{\theta}} [\|\nabla z_i(x)\|_2 \, \rho_{b_i}(z_i(x))] \quad (39)$$

Using this we can also get the estimate of the local complexity density function $f$:

$$f(x) \leq \sum_{\text{neuron } z_i} \mathbb{E}_{\tilde{\theta}} [\|\nabla z_i(x)\|_2 \, \rho_{b_i}(z_i(x))] \quad (40)$$

We can now empirically estimate the right hand side of (40) by using computing finite samples of perturbations to the biases and taking the empirical mean. In particular we use $\sigma = 0.05$ for Figure 1. We provide here an ablation on the choice of $\sigma$ in Figure 4. In Figure 5 we provide an example illustrating the effect of adding noise not only to the biases but also to the weights.

Several of our bounds, in particular those derived from Corollary 3, rely on removing the term $\rho_b$ from the summand when computing the Local Complexity. We demonstrate in Figure 6, the effect of estimating the local complexity density function as:

$$\hat{f}(x) = \sum_{z \text{neuron}} \mathbb{E}_{\tilde{\theta}} [\|\nabla(x)\|_2]. \quad (41)$$

We show in Figure 6 what this density function looks like, and we can see that it still bears a strong qualitative resemblance to the original structure of linear regions from Figure 1.

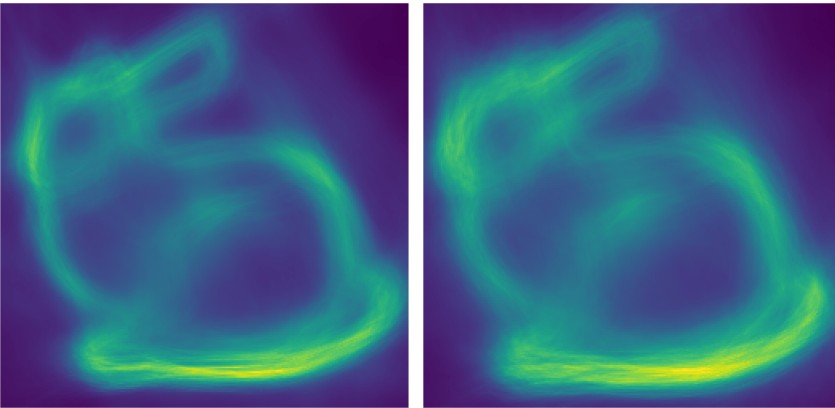

Figure 5: Here we plot the local complexity density function $f$ comparing the effects of adding noise to just the biases (Left) vs adding the same amount of noise to both the biases and the weights (Right). Here we used $\sigma = 0.05$. As we see, the effects are qualitatively similar in both cases.

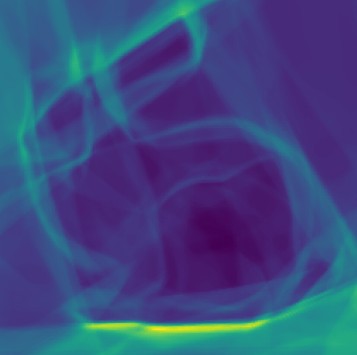

Figure 6: Effect of plotting $\hat{f}$ as defined in (41). The setup is otherwise the same as that in Figure 1 and Figure 4.

## B.2 DETAILS ON FIGURE 2

We create a synthetic dataset by sampling from an isotropic Gaussian $X$, and a correlated isotropic Gaussian $Y$. The cross-covariance matrix between $X$ and $Y$ is randomly generated. In these examples we use an input dimension of $100$ and an output dimension of $2$. We train with the Adam optimizer with learning rate $1e - 4$. We show that this effect is the same across several training runs, each with a different cross-covariance matrix in Figure 7.

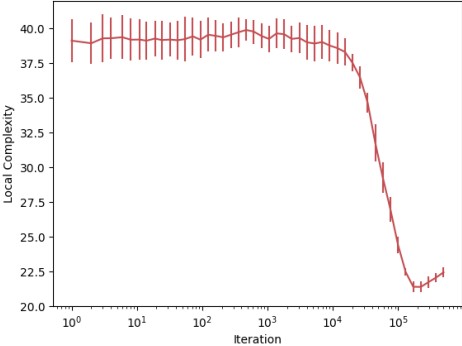 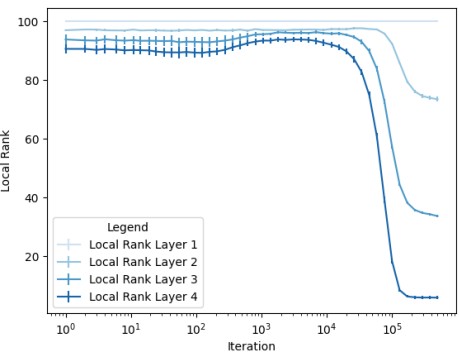

Figure 7: Here we run the same experiment as in Figure 2, 6 times, each with a different cross-covariance matrix. We demonstrate that this effect is consistent by plotting standard deviation error bars on each collected data point. We find a Pearson's correlation coefficient of $0.852$ between the local complexity and the local rank at layer 2, and a Pearson's correlation coefficient of $0.957$ and $0.985$ at layers 3 and 4 respectfully.

## B.3 MORE INFORMATION ON FIGURE 3

Here we compute the local complexity as in Figure B.1 by computing the gradients at each neuron $\nabla z(x)$ and computing a mean over data points in the test dataset. Similarly, we estimate the total variation of our network by computing the mean of $\|\nabla \mathcal{N}(x)\|$ at points in the test dataset.

We note that we see most clearly the relationship between the total variation and the local complexity when training with a high initialization scale. In Figure 3 we initialize our weights with a standard deviation twice that of the typical He initialization scheme (He et al., 2015). This approach is commonly employed in the literature when investigating grokking and the terminal phase of training (Fan et al., 2024) (Lyu et al., 2024). Nevertheless, in Figure 8 we perform an ablation study on the initialization scale. In both cases we can see an increase in the adversarial accuracy late in training corresponding to a drop in the local complexity, but the correlation between the local complexity and the total variation seems to break down at lower initialization scales. So, our theoretical works appear to not fully describe the dynamics in certain cases.

## B.4 REMARKS ON TIGHTNESS OF THE BOUNDS

We observe in Figure 8 that the total variation occasionally fails to decrease alongside the local complexity, which raises questions about the tightness of the bound in Theorem 7. While the exact relationship between total variation and local complexity is complex, these empirical findings do not necessarily invalidate the bound. The bound as stated depends on the term $\max_{1 \le l \le L} C_l$, where $C_l$ represents the Lipschitz constant of $g_l$ (the rest of the network following the $\bar{l}$-th layer). To empirically verify this bound's validity, we need to compute or estimate this term. We propose the following crude approach for estimating the Lipschitz constant term:

$$\max_{1 \le l \le L} C_l \le \max_{1 \le l \le L} \|W_l W_{l+1} \cdots W_L\|_{op}.$$

The above inequality is tight if there is a linear regions for which all neurons are active. Using this as an estimate for $\max_{1 \le l \le L} C_l$, we can then compute an empirical estimate for the term:

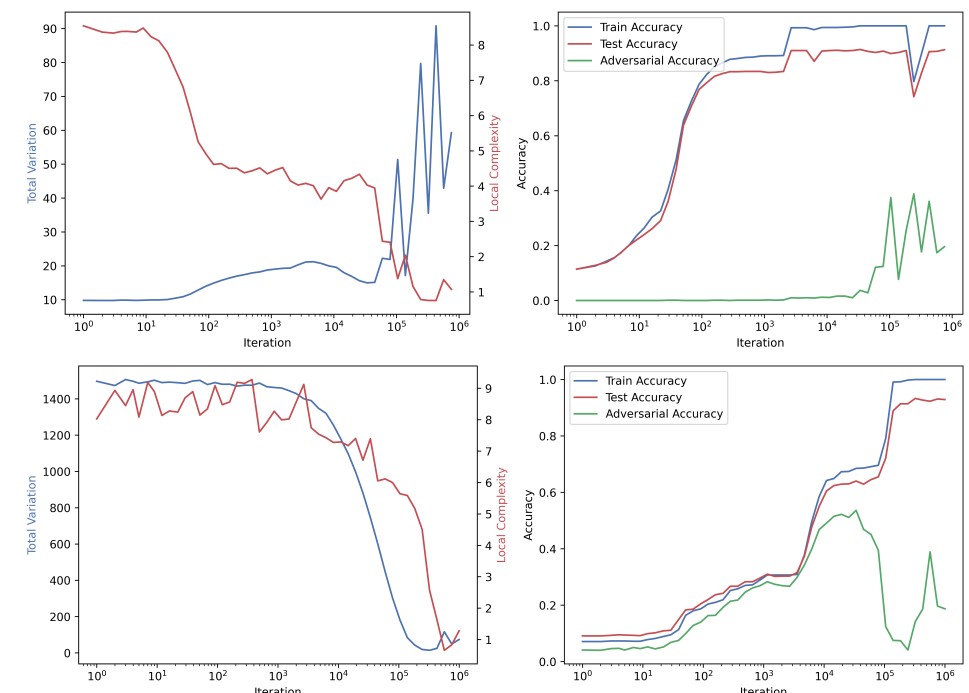

Figure 8: Here we demonstrate the results of training an MLP on a subset of the MNIST dataset with the standard He initialization (Top) and $3x$ the regular He intialization. This model has the same architecture as that in Figure 3.

$$\mathbf{TV} \cdot \frac{Lc_{\text{bias}}^{\eta}}{\max_{1 \leq l \leq L} C_l} \approx \mathbf{TV} \cdot \frac{Lc_{\text{bias}}^{\eta}}{\max_{1 \leq l \leq L} \|W_l W_{l+1} \cdots W_L\|_{op}}. \tag{42}$$

We visualize the relationship between this quantity and the Local Complexity in Figure 9. When comparing Equation (42) with the local complexity, we find that the observed increases in total variation during late-stage training can be attributable to larger Lipschitz constants $C_l$, rather than an inherent looseness in the bound. This observation suggests further intriguing and unexpected behavior during the terminal phase of training that merits further investigation.

### ON THE NUMBER OF NEURONS WHICH ARE NOT GOOD

Many of our lower bounds also involve a factor $B$, which we define to be the expected number of neurons which are not good when evaluated over the data distribution. In particular, we will measure,

$$B = \mathbb{E}_{x \sim p} \left[ \sum_{\text{neuron } z_i} \mathbb{1}_{z_i \text{ not good at } x} \right].$$

For a fully connected network, a neuron would be not good at $x$ only if there is a layer in the network for which every neuron is off when evaluated at $x$. This means that this quantity would be quite small for networks of reasonable width, as we can see in Figure 10.

## C ADDITIONAL FIGURES

### C.1 CLUSTERS IN WEIGHT VECTORS AFTER DROP IN LOCAL COMPLEXITY

Here, in Figure 11, we demonstrate the emergence of structure in UMAP plots of weight vectors late in training. This connects to the concept that, in the kernel regime, networks fit data points without

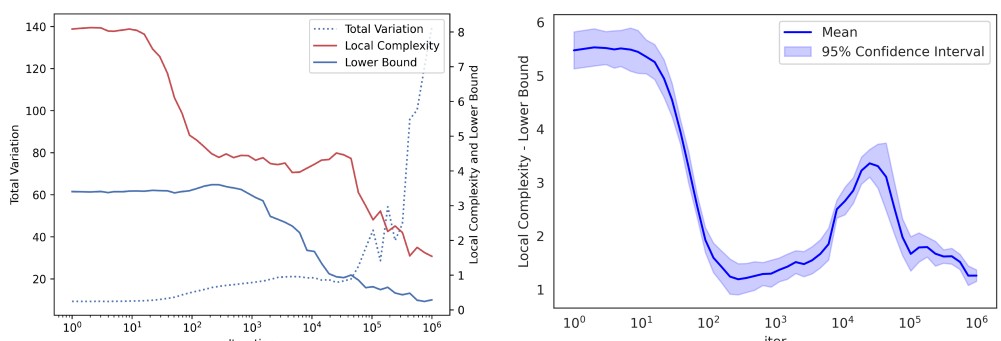

Figure 9: Here we train a network on MNIST with the He initialization scheme, 4 hidden layers each with dimension 200. We see a spike in the Total Variation late in trainig (dotted line). On the left, can also see that the lower bound as estimated via equation (42) still decreases along with the local complexity in the terminal phase of training. On the right, we show that this behavior is reproducible by running the same experiment 8 times and computing a confidence interval of the term $\mathbf{LC} - \frac{\mathbf{TV} \, Lc_{\mathrm{bias}}^{\eta}}{\max_{1 \leq l \leq L} C_l}$ We use a $\eta = 1$, $\sigma = 1$, to estimate the constant terms, as we find that this choice of $\eta$ maximizes the tightness of the lower bound from 7.

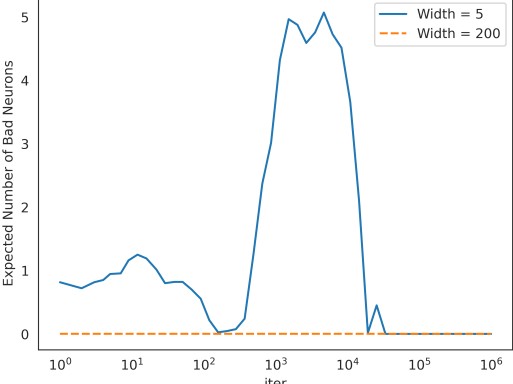

Figure 10: Here, we plot the empirically observed value of the number of neurons which are not good, $B$, for an MLP during training on MNIST. Both networks have depth 4, and we can see that for the wider network $B = 0$ at all timesteps.

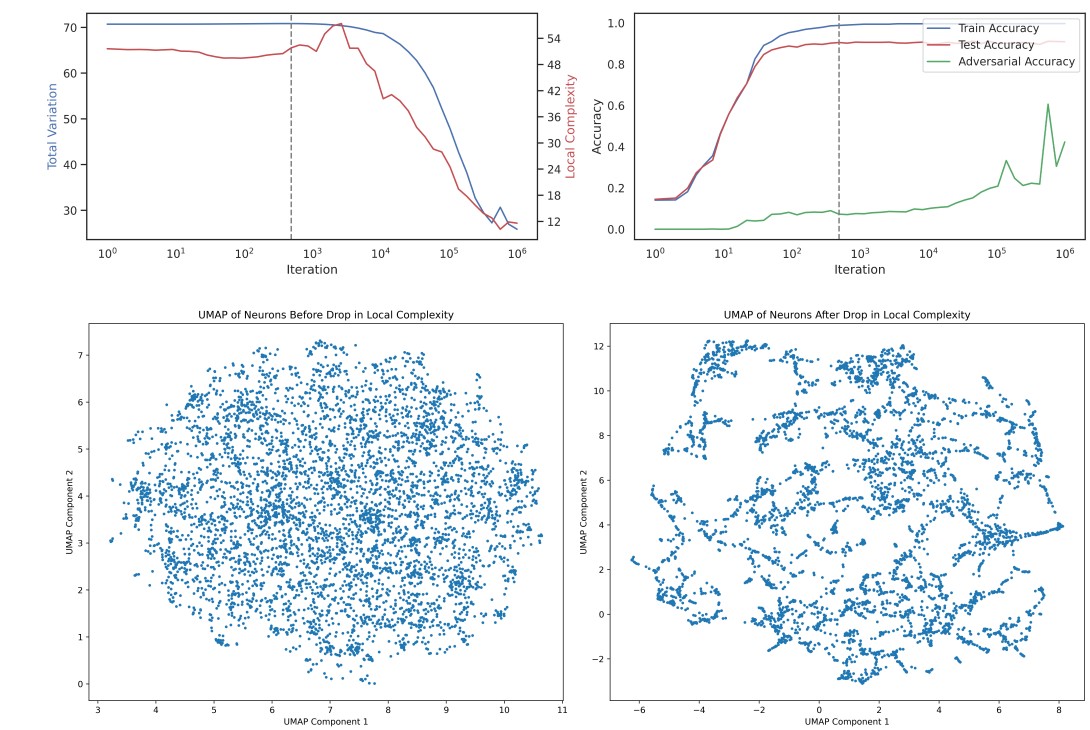

Figure 11: Here we demonstrate qualitative changes in the parameters before and after the drop in local complexity. We consider here a one hidden layer MLP trained on a subset of 1000 images of the MNIST dataset. The hidden layer has 5000 neurons. We plot a low-dimensional UMAP visualizations of the weight vectors associated to each neuron in the hidden layer at 494 iterations (marked by dashed line) and at $1,000,000$ iterations.

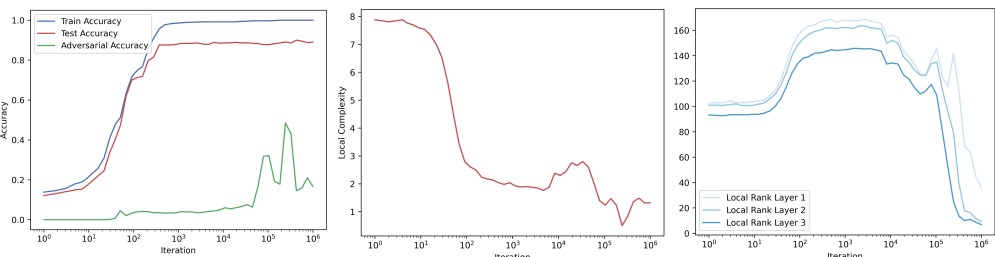

Figure 12: Local Rank Analysis on the MNIST Dataset. In this figure we train an MLP on MNIST with 3 hidden layers of 200 neurons each. We use a regular 1x initialization scale.

substantially altering the structure of their linear regions. However, after transitioning to the rich training regime, we observe more intricate clustering in the weight vectors, providing evidence of feature learning.

## C.2 LOCAL RANK ON MNIST

In Figure 12 we demonstrate the dynamics of Local Rank when training with a subset of 1000 images of the MNIST dataset. We note that the drop in the local rank approximately corresponds to the second drop in the local complexity, as well as the increase in the adversarial robustness of the network.

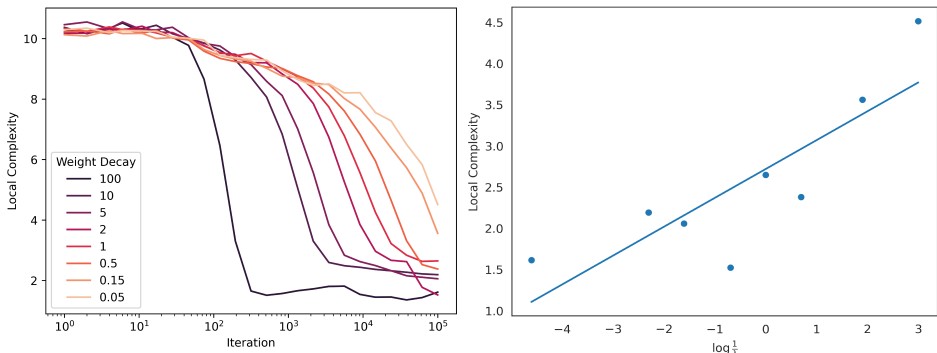

Figure 13: Here we demonstrate a correlation between the weight decay parameter and the drop in local complexity late in training. On the left, we note that this drop appears to come earlier for higher values of the weight decay parameter. On the right, we plot the bounding quantity from (17) on the $x$-axis, and the local complexity at the end of training on the $y$-axis. We also plot a linear regression, and observe an $R^2 = 0.6972$. In these experiments, we consider a shallow 2 layer MLP, with a hidden-layer dimension of 1000. This network is trained on a subset of MNIST with the Adam optimizer and learning rate $1e-4$.

### C.3 LOCAL COMPLEXITY AND WEIGHT DECAY

In Figure 13 we demonstrate a correlation between the weight decay parameter over several training runs, and the drop in local complexity late in training, which relates to our results in Proposition 10.

