# OpenReview forum: "On the Local Complexity of Linear Regions in Deep ReLU Networks"
_ICLR.cc/2025/Conference — Submitted to ICLR 2025_

### Official Review · Reviewer_FXxS · 2024-10-28

**Soundness:** 1
**Presentation:** 2
**Contribution:** 1
**Rating:** 5
**Confidence:** 3

**Summary:**

The authors define a novel notion of complexity for functions in relation to some data distribution, they call it Local Complexity. Local Complexity measures the average "density" of discontinuities in the gradient of the function in the input space. For ReLU networks, this translates to the number of linear regions under mild assumptions.

The authors show that Local Complexity is closely related to the notion of Local Rank.

The authors bound a notion of Total Variation of functions by their Local Complexity, thereby getting some proxy for the susceptibility of the function to adversarial examples, and whether "mode collapse" has happened during training.

The authors bound the Local Complexity of neural networks by their representational costs - the minmal norm of weights realizing the same networks, to conclude that weight decay and other forms of norm minimization lead to low Local Complexity and small number of linear regions which explains empirical observations.

Finally, they bound the Local Complexity at later, "rich" phases of training.

**Strengths:**

The paper presents a new measure of complexity for functions, connects it to different measures that haven been researched, and sheds some light on / provides directions as to how this measure might shed light on interesting open questions in the field.

**Weaknesses:**

I would like clarifications as of the significance of the notion of Local Rank. Cited works in the "Related Works" section mention roughly similar definitions, and the cited work (Chechik et al.) does not mention the term local rank, and discusses ranks of matrix transformations between gaussian variables, which seems unrelated.

In the section on Total Variation provides a theoretical hint w.r.t. the relationship between LC and TV, but as the authors point - it is insufficient. On the other hand the empirical experiments provided in the appendix are less than sufficient, and even show a single case of the bound's failure under the standard initialization. Please provide more significant empirical evidence for the phenomenon, or a sound proof.
Also, please provide a (not intuition based) explanation for the connection between TV and adversarial robustness.

In the section about LC and optimization, clarify the significance of the bound in proposition 8. The dependence on the norm-relations seems to be additive, while the other term seems to be exponentially dependent on the depth. This I think could easily be looser than the bounds in Hanin et al., for example, which as far as I can tell use a measure that upper bounds the LC.
As of corollary 9 - there is only one experiment in the appendix, which would enjoy the benefit of plotting the bounding quantity at hand ($\Theta(\log(1/\lambda))$.

**Questions:**

1. What is the significance of the notion of Local Rank? There are other notions of rank that are cited to have some results tied to them, but how is the LC connected to those?
2. How does the bound in proposition 8 fare against similar bounds (i.e. of Hanin et al.). I understand the setting is a bit different, and yet it is unclear whether this bound is vacuous or not.
3. What is the empirical\formal connection between Total Variation and Adversarial Robustness? Excluding of the intuitive explanation that is provided in the paper.

---

> ### Author Response · Authors · 2024-11-22
> **Part I**
>
> We sincerely thank the reviewer for their thoughtful and detailed comments which have helped us improve our paper. Below we address the items raised in the review.
>
>
> > What is the significance of the notion of Local Rank?
>
> The relevance of the Local Rank is that it is a natural concept that we can relate to other interesting concepts. In our work, we related the Local Complexity and the Local Rank.
> We have updated the mentioned reference (Chechik et al.) with the more relevant (Patel \& Shwartz-Ziv, 2024).
> The Local Rank is a definition that is closely related to ideas in differential geometry, particularly the rank of a smooth map, submersions and immersions of manifolds.
> Since this gives a way to quantify the rank in a non-linear manner, we think that this is a natural definition.
> It is also related to other notions of rank that appear in the literature. Some works look at the rank of the feature matrix, which is the $d\times n$ matrix of $d$-dimensional latent features for $n$-samples. If this matrix has rank $r < \min \{n,d\}$, then this means that the network maps input points to a low dimensional subspace.
> This also means that the feature manifold must have rank less than or equal to $r$, so the rank of the embedding matrix bounds the Local Rank.
> Another relevant way rank appears is with respect to the rank of weight matrices.
> If we suppose that the $l$-th layer weight matrix has rank $r$, then the Local Rank must be less than or equal to $r$, since we can compute that the Jacobian of the network to the $l$-th layer is $W_1 D_1 W_2 D_2 \cdots D_{l-1}W_l$, where $D_i$ are diagonal $0,1$ matrices corresponding to the activation patterns of the ReLU functions at the $i$-th layer.
>
> > How does the bound in proposition 8 fare against similar bounds (i.e. of Hanin et al.).
>
> Our work develops a theoretical framework that shares similarities with the approach developed by Hanin \& Rolnick to study linear regions, particularly in the application of the co-area formula. However, the bound in Proposition 8 (now Proposition 9 in the updated draft) and the bounds of Hanin \& Rolnick are describing different quantities and as such are not comparable. The result in our proposition provides a view into how the local complexity relates to works on implicit regularization.
> This allows us to understand how the linear regions may look like after training.
> On the other hand, the bounds by Hanin \& Rolnick describe the expectation value of the number of linear regions over the input space given a distribution over the parameters. Their work does not pursue a description of the distribution of the linear regions for a specific choice of parameters nor provides results about how the linear regions may look like after training.

---

> > ### Author Response · Authors · 2024-11-22
> > **Part II**
> >
> > > On the other hand the empirical experiments provided in the appendix are less than sufficient, and even show a single case of the bound’s failure under the standard initialization. Please provide more significant empirical evidence for the phenomenon, or a sound proof.
> >
> > We have revised the experiments and we have added a new section (Appendix B.4) analyzing empirically the tightness on the bound of the Total Variation.
> > We find that the observed increase in total variation late in training for some initializations is primarily driven by an increase in the Lipschitz constants of the layer-wise functions $g_l$, rather than a fundamental looseness in our bounds.
> > We have empirically measured the Lipschitz constants during training, showing that the ratio between our theoretical upper bound and the observed values remains relatively stable during training.
> > Previously we had reported an increase in the total variation late in training for some initialization scales. We observe that the theoretical result is a bound for the ratio of the Total Variation and a maximal Lipschitz constant of a subset of the layers. When this is considered, the theoretical bound tracks the experimental results as shown in Figure 9. We reproduced this experiment over several training runs and show that this result is consistent.
> >
> > In the present work we aim to provide a theoretical framework to establish links between the linear regions in ReLU networks and other relevant geometric properties. Our motivation is to complement recent experimental works from a theoretical perspective.
> > We agree that experimental analysis is valuable and that is why we also included some experimental evaluations in our work, which we will continue to expand and update in this line of investigation. At any rate we would direct to the work of Humayun et al. (2024) for a more detailed empirical study of the relations between the linear regions and adversarial robustness, including an examination across different scenarios and architectures.
> >
> >
> > We would also like to note that the bounds relating the local complexity and the rank of learned representations in Section 4 are controlled in the sense that we have both an upper and lower bound connecting the local complexity to the local rank. See Theorem 5, Equations (11) and (12).
> >
> > > What is the empirical / formal connection between Total Variation and Adversarial Robustness
> >
> > We have added Proposition 6, which relates the total variation and the existence of adversarial examples. Here we consider a simplified setting that allows us to show how these quantities are intrinsically related. We are able to show that a low total variation implies that adversarial examples will not exist near data points.
> >
> > > As of corollary 9 - there is only one experiment in the appendix, which would enjoy the benefit of plotting the bounding quantity at hand
> >
> > We thank the reviewers for this note. We have revised the figure to include a direct comparison between the bounding quantity $\log \frac 1 \lambda$ and the local complexity at the end of training. We observe that there is a linear relationship between these quantities with an $R^2 =  0.6972$. This means that $69.72$% of the variance in the local complexity at the end of training can be explained by the term $\log \frac 1 \lambda$.
> >
> > Please let us know if any aspects of our work would benefit from further clarification.

---

> ### Author Response · Authors · 2024-11-27
>
> As the author-reviewer discussion period is ending soon, we kindly ask if you could review our responses to your comments. If you have further questions or comments, we will do our best to address them before the discussion period ends. If our responses have resolved your concerns, we would greatly appreciate it if you could update your evaluation of our work accordingly.

---

### Official Review · Reviewer_zmzH · 2024-10-29

**Soundness:** 3
**Presentation:** 3
**Contribution:** 3
**Rating:** 6
**Confidence:** 3

**Summary:**

The paper introduces the concept of local complexity in neural networks with piecewise linear activations (specifically ReLU networks) as a measure of the density of linear regions over input data distributions. The authors demonstrate the connections between local complexity and total variation, adversarial robustness, and representation learning. The work contributes a theoretical framework relating geometric properties of ReLU networks, specifically the linear regions, to different aspects of learning.

**Strengths:**

- The paper presents a mathematical foundation for the definitions and claims of local complexity, including theorems and proofs that enhance the credibility of the findings.
- The links drawn between local complexity, feature learning, and robustness are timely and relevant, aiming at understanding important aspects of deep learning.
- Also, the empirical results help support the theoretical claims, providing a practical understanding of the phenomena discussed.

**Weaknesses:**

- Definitions and Assumptions: The definitions of local complexity, local rank, and good neurons, while mathematically interesting, maybe too complex for practical application. A clearer explanation or simplification of these concepts would aid in understanding their implications for practitioners. Some sections of the paper are dense and may be challenging for readers unfamiliar with the topic. Improving the clarity and flow, particularly in the introduction and theoretical results, would enhance readability.

- Empirical Validation: the paper could benefit from more extensive empirical experiments to validate the claims regarding local complexity and its relationship with representation learning and adversarial robustness. Particularly those concerning adversarial robustness might be sensitive to specific training conditions or datasets. A broader analysis across different scenarios would be necessary to ensure that the findings are not artifacts of particular experimental setups.

- On Practical Implications: The paper lacks a thorough discussion of the practical implications of the findings for model design and training. Insights into how practitioners can leverage local complexity in their work would be valuable.

**Questions:**

- Regarding the definition of local complexity: can authors provide more intuitive explanations on why the noisy biases are considered instead of noisy weights? Would the local complexity defined with respect to noisy weights lead to similar subsequent results? This can be a fundamental question as weights can affect the linear regions in a more complex way.

- Regarding Corollary 3: can authors provide more explanations on the defined constants $c_{bias}^\eta$ and $\bar{\xi}_{\eta}$? What is the meaning of these quantities and how are they related to the local complexity? Also, the definition of $\Theta$ seems missing.

- Regarding Theorem 6 and Proposition 7,8: How do these results directly relate to existing theories on model complexity and adversarial robustness? Are there specific studies that the authors see as complementary or contradictory to their findings?

- Regarding the applications: How can practitioners leverage the findings on local complexity in their model design and training processes? What specific recommendations can the authors provide based on their results?

---

> ### Author Response · Authors · 2024-11-22
> **Part I**
>
> We sincerely thank the reviewer for their thoughtful and detailed comments which have helped us improve our paper.
> Below we address the items raised in the review.
>
> > A broader analysis across different scenarios would be necessary to ensure that the findings are not artifacts of particular experimental setups.
>
> In the present work we aim to provide a theoretical framework to establish links between the linear regions in ReLU networks and other relevant geometric properties. Our motivation is to complement recent experimental works from a theoretical perspective.
> We agree that experimental analysis is valuable and that is why we also included some experimental evaluations in our work, which we will continue to expand and update in this line of investigation. At any rate we would direct to the work of Humayun et al. (2024) for a more detailed empirical study of the relations between the linear regions and adversarial robustness, including an examination across different scenarios and architectures.
>
> > can authors provide more intuitive explanations on why the noisy biases are considered instead of noisy weights?
>
> We have revised Section 3.1 to better explain our approach.
> Our theoretical framework builds on the analysis of the threshold at which a neuron transitions from inactive to active.
> The biases directly control the level at which a ReLU transitions from being inactive to being active and that is why we focus on bias perturbations.
> Perturbations to the weight matrix introduce changes in the direction of these boundaries and are naturally more complex. However, we can also include perturbations of the weights and extend all definitions to also include perturbations of the weights.
> We have added Figure 5 in the appendix to illustrate how adding perturbations to the weight matrices does not qualitatively change our measure of the local complexity.
>
> > Regarding Corollary 3: can authors provide more explanations on the defined constants $c^\eta_\text{bias}$ and $\bar \xi_\eta$? What is the meaning of these quantities and how are they related to the local complexity?
>
> These constants give us a trade-off parameterized by $\eta$ when computing a lower bound on the local complexity, which controls the tightness of the bound.
> In our proof, these constants serve to bound separately the cases where the noisy biases fall too far away from their mean via concentration inequalities.
> This trade-off can also be represented in the form $c^\eta_\text{bias} (\mathbb E\|\nabla z(x)\| - \Theta(\frac1{\eta^2}))$.
> We can tell that this never trivializes our bound for some choice of $\eta$, as $c^\eta_\text{bias} = \frac1{\sqrt{2\pi}\sigma} e^{\frac{\eta^2}{2\sigma^2}}$ is always positive.
> So we can choose $\eta$ large enough such that $\mathbb E\|\nabla z(x)\| > \frac{1}{\eta^2}$, and we get that the whole term $c^\eta_\text{bias} (\mathbb E\|\nabla z(x)\| - \Theta(\frac1{\eta^2})) > 0$, so this bound is not trivial.
>
> > Also, the definition of $\Theta$ seems missing.
>
> We have added a footnote on page 6 that explains our use of the notation $\Theta$, which refers to the standard asymptotic notation: $f(x) = \Theta(g(x))$ if $f$ is asymptotically bounded above and below by $g$, meaning that there exist $c_1, c_2, x_0$ such that for any $x > x_0$, $c_1g(x) \leq f(x) \leq c_2g(x)$.
>
> > How do these results directly relate to existing theories on model complexity and adversarial robustness?
>
> We think that the closest comparison of our work is the empirical study of Humayun et al. (2024). Our work complements this work by providing a theoretical framework and theoretical results. Other related works include that of Li et al. (2022), which relates adversarial robustness to the VC dimension of the function class. This is similar in flavor to our results, as they show for small enough VC dimension, one can prove robustness results. Another result that links complexity of the linear regions to robustness is that of Humayun et al. (2023), which leverages the linear region structure of ReLU networks to design an algorithm which improves adversarial robustness. Croce et al. (2019) is another relevant work, which relates the size of linear regions to adversarial robustness, and provides an algorithm which increases the size of linear regions in a way to try and improve robustness. We have added some of the missing references to the Related Works section to provide a better overview of the field. Please let us know if you are aware of any other relevant references.

---

> > ### Comment · Reviewer_zmzH · 2024-11-23
> >
> > I thank the authors for the detailed response. My concerns are partially addressed. While I am not very convinced by the answers regarding the noisy weights and Corollary 3.
> >
> > - Based on Figure 5 in the appendix, adding perturbations to **both** the weight matrices and bias vectors seems not change the measure of the local complexity. defined by adding perturbations to bias vectors. Would that be different if simply adding perturbations to the weight matrices but not to bias vectors? In other words, in Figure 5, the effects of noisy weights may not be truly revealed given both experiments are conducted with noisy bias.
> >
> > - Regarding Corollary 3, with large enough $\eta$, the constant $c^\eta_{bias}=\frac{1}{2\pi}\exp(-\eta^2/2\sigma^2)$ also approaches to zero, learning to a vanishing lower bound. The right-hand side in (8) can be negative, which leads to a trivial lower bound. Does that make sense?

---

> > > ### Author Response · Authors · 2024-11-26
> > >
> > > Thank you for your ongoing conversation and review of our work. We address your questions below:
> > >
> > > 1) Our theory centers around taking the expectation of a random level set of a function. This is illustrated in the proof of Lemma 12.
> > > As indicated in our previous responses, perturbations of the biases control the level sets directly which makes the resulting formulas more tractable.
> > > Taking perturbations only for the weight matrices and leaving the biases fixed is an interesting idea, which however will lead to formulas that are less tractable.
> > > Note in particular that in the case of fixed biases the results of Hanin and Rolnick (2019) will also fail, since their results require that the biases admit a density when conditioned on all the other weights.
> > >
> > > 2) We also note that the bound in question holds for any choice of $\eta$. We would generally want to pick an $\eta$ such that this bound is as tight as possible. We can do this by solving for the maximum of an equation which looks roughly like the following (we will take $C =\mathbb E|\nabla z|$ here and simplify the constant factors),
> > > $$
> > > g(\eta) = e^{-\eta^2} (C - \frac{1}{\eta^2}).
> > > $$
> > > This has an positive optimum at
> > > $$\eta_* = \sqrt{\frac{1+\sqrt{1+4C}}{2C}}.
> > > $$
> > > We think that this is not too loose of a bound, especially for large values of $C$. Here, one could see that $g(\eta_*)/C \to 1$ as $C\to \infty$.
> > >
> > > We note that the bound also involves a constant $B$. As discussed in the text, for wide networks one can show that this is very small at initialization. In practice this also remains small during training. Note that in a fully-connected network, for a single neuron to be not good at $x$, we would require that there is some layer where every neuron in the layer is inactive at $x$. In our new experiments in Appendix B.4, this does not happen for any sampled $x$ during training for networks of reasonable width. We have added a note in the main text to reflect this.

---

> > > > ### Comment · Reviewer_zmzH · 2024-11-27
> > > >
> > > > I thank the authors for the further response. They partially addressed my concerns. I would retain my original score for being on the acceptance side.

---

> ### Author Response · Authors · 2024-11-22
> **Part II**
>
> > How can practitioners leverage the findings on local complexity in their model design and training processes?
>
> Our investigation leads to several actionable insights. For one, we demonstrate that robustness and the local complexity are related concepts. Following this, to improve robustness one would want to design the optimization procedure in a way that minimizes (explicitly or implicitly) the local complexity.
> We show that training in the kernel regime leads to limited change in the local complexity.
> In view of this, initialization and learning rate schemes that promote feature learning, and leave the kernel regime, would then have positive effects on the robustness of trained networks.
> We also provide, in Section 6.2 and Figure 12, a connection between the use of weight decay and the local complexity, showing that large values of the weight decay parameter may implicitly regularize the local complexity.
>
> We have also made additional improvements to strengthen the paper in this regard. These include adding Proposition 6, which directly relates the total variation and the existence of adversarial examples in a simplified setting. We have also added Appendix A.10, which bounds the change in the local complexity when the change in the parameters is small. Please let us know if any aspects of our work would benefit from further clarification.

---

### Official Review · Reviewer_DUki · 2024-11-03

**Soundness:** 3
**Presentation:** 3
**Contribution:** 3
**Rating:** 6
**Confidence:** 3

**Summary:**

The paper studies the concept of "local complexity" in deep ReLU networks, defining it as the density of linear regions. This complexity metric relates to representation learning, robustness, and parameter optimization. The authors show that networks that learn lower-dimensional features exhibit lower local complexity, which correlates with reduced total variation and improved robustness.

**Strengths:**

**Clarity and Quality**

Very well-written and clearly structured paper. The problem has been motivated nicely in the introduction. The related work section is rigorous and clearly details the current results for this problem.


**Theoretical Insights**

By linking local complexity to local rank (feature learning), total variation (robustness), and representation cost (parameter optimization), the authors build a solid theoretical foundation that advances the understanding of ReLU networks’ geometric properties.
Also, they empirically demonstrate such relations by conducting several experiments.

**Weaknesses:**

**Originality**

- The authors define the local complexity of networks inspired by (Hanin & Rolnick, 2019b). However, there is no direct comparison between local complexity with complexity defined by (Hanin & Rolnick, 2019b) though definitions are quite similar.  I think this makes the contribution of this paper incremental. To highlight the novelty, it would be better to introduce the main difference.

e.g. Why local complexity is much more appropriate for analyzing local rank, total variation, and representation cost?


**Tightness**

As illustrated in the paper, relations between local complexity and others can be loose (i.e., loose inequality). Nonetheless, if such relations cannot be characterized by complexity defined by (Hanin & Rolnick, 2019b), it would be not a weakness. Thus, as explained above, please
introduce the main difference.

**Questions:**

**Non-differentiable**

Though ReLU networks are non-differentiable at $0$, there is no explanation for it. Is every gradient in the paper subderivative?
If so, are several definitions (e.g. total variation) well-defined? For instance, to define total variation using a gradient, the function $f$ should be differentiable.

**Piecewise linear activation**

The complexity of linear regions in networks using piecewise linear activation is often studied.[Hanin & Rolnick, 2019].
Can we get similar results (thm. 5~ Prop.8) for piecewise linear activation?

---

I am hoping that the authors will provide the clarifications stated therein in the rebuttal phase

---

> ### Author Response · Authors · 2024-11-22
>
> We sincerely thank the reviewer for their thoughtful and detailed comments which have helped us improve our paper.
> Below we address the items raised in the review.
>
> > (Originality)
>
> Our theoretical framework shares similarities with that of Hanin \& Rolnick, particularly in the application of the co-area formula to discuss linear regions.
> However, our definitions and results differ in several key aspects.
> Hanin \& Rolnick investigate how the depth and number of neurons affect the expected number of linear regions for a generic distribution of parameters.
> In contrast, in our definitions and results we are concerned with how the linear regions vary across different regions of parameters and we are concerned with the distribution of linear regions over different portions of the input space, which are aspects that are not discussed in the work of Hanin \& Rolnick.
> These differences are reflected in our definitions, where we consider perturbations around a specific parameter and we consider the density of linear regions over the input space integrated over an input data distribution.
> We may also point out that the paper of Hanin \& Rolnick does not pursue results describing properties such as the Local Rank or the Total Variation, which we investigate in our work.
> We have added a note in Section 3.1 in the revision to further highlight these differences.
>
> > (Tightness)
>
> Regarding the tightness of theoretical bounds, we have added a new section (Appendix B.4) analyzing empirically the tightness on the bound of the Total Variation.
> We find that the observed increase in total variation late in training for some initializations is primarily driven by an increase in the Lipschitz constants of the layer-wise functions $g_l$, rather than a fundamental looseness in our bounds.
> We have empirically measured the Lipschitz constants during training, showing that the ratio between our theoretical upper bound and the observed values remains relatively stable during training.
> Previously we had reported an increase in the total variation late in training for some initialization scales, but observe that the theoretical result is a bound for the ratio of the Total Variation and a maximal Lipschitz constant of a subset of the layers. When this is considered, the theoretical bound tracks the experimental results as shown in Figure 9.
>
>
> We would also like to note that the bounds relating the local complexity and the rank of learned representations in Section 4 are controlled in the sense that we have both an upper and lower bound connecting the local complexity to the local rank. See Theorem 5, Equations (11) and (12).
>
> > (Non-differentiable)
>
> It is correct that ReLUs are not differentiable at 0.
> In our results we use the derivatives of neurons with respect to the input to the network. These functions are almost everywhere differentiable over the input space and since our results integrate over an input density, the non-differentiable points are inconsequential and have no bearing on the results.
> We have added a clarification to the text in Section 3.2.
>
> We would also like to note that the total variation is well defined for many functions which do not admit a derivative. In particular, it can be defined for any $L^1$ integrable function.
>
> > (Piecewise Linear Activation)
>
> In the present work we focus on the ReLU activation as this is the most commonly used piecewise linear activation. However, we believe that similar results should also hold for other piecewise linear activation functions. A program in that direction could follow some of the strategies from the work of Tseran and Montufar (2021) on the expected number of linear regions for maxout networks. This would be an interesting and natural generalization. We have added a note on this in the conclusion.
>
> > (Other)
>
> As it may be of interest, we have added Proposition 6, which relates the total variation and the existence of adversarial examples in a simplified setting. We have also added a Appendix A.10, which bounds the change in the local complexity when the change in the parameters is small.
> Please let us know if any aspects would benefit from further clarification.

---

> > ### Comment · Reviewer_DUki · 2024-11-26
> >
> > Dear authors,
> >
> > I thank the authors for their detailed response and for addressing my concerns. I would retain my original score.

---

### Official Review · Reviewer_iFkq · 2024-11-03

**Soundness:** 3
**Presentation:** 3
**Contribution:** 3
**Rating:** 6
**Confidence:** 3

**Summary:**

This work defines the local complexity of a ReLU network and demonstrates the link between the complexity of linear regions and adversarial robustness. The local complexity is shown (roughly) equal to the expected input-gradient norm of neurons. Adversarial robustness relates to the total variation of a network, which is defined as the input-gradient norm of the network and thus can be linked to the local complexity. Empirically, it has been observed that the adversarial robustness shows a sharp increase during training when the ReLU network becomes geometrically simple in the input space (e.g., fewer linear regions). The authors confirm by numerical experiments that the sharp increase in adversarial robustness correlates with local complexity and rank.

**Strengths:**

- This study presents a rigorous connection between the geometric characteristics of the ReLU network and adversarial robustness.
- The theoretical analysis links local complexity to the local rank, which relates to the input-space geometry of the ReLU network, and to the total variation of the ReLU network, which relates to the adversarial robustness, thereby linking the linear regions and adversarial robustness.

**Weaknesses:**

I raise the following as the major weaknesses of this work.
1. The analysis of the time evolution of the local complexity and adversarial robustness is limited.
2. The tightness of the bound is not discussed by theory or experiments.

I elaborate on the weaknesses below.

1. The time evolution analysis
As given in the introduction and related work, this study is interested in the observation of the sharp increase in adversarial robustness and more straightforward linear regions in the late stage of the training.
The authors provide the link between the robustness and the simplicity of the linear region but do not show their time evolution. Thus, no comprehensive answer to the motivating observation is provided.
Proposition 8 and Corollary 8 give upper bounds for the local complexity in the rich regime, but there is no clarification about the kernel regime. Is there any lower bound for the local complexity in the kernel regime?

2. The tightness of the bound.
This paper provides several bounds, such as Eq. (11), (12), and (13), but it is not clear how tight these bounds are. It may be theoretically hard, but at least it is doable to show it by comparing theoretical and empirical plots.

***Minor comments***
- [Below Eq. 17] Appendix 10 was not found (and it seems not the typo of Figure 10).

**Questions:**

Please answer the two weaknesses raised above.

---

> ### Author Response · Authors · 2024-11-22
>
> We sincerely thank the reviewer for their thoughtful and detailed comments which have helped us improve our paper. Below we address the items raised in the review:
>
> > Is there any lower bound for the local complexity in the kernel regime?
>
> In the kernel regime the weights do not move far from their initial values during training. Thus, in this setting, we expect that results that hold at initialization should also approximately hold after training.
> We have added a theoretical result (Proposition 17 in Appendix A.10) that shows this for a simple setting.
> Here we show for two layer networks that, when the parameters stay close to initialization, the local complexity stays close to the local complexity at initialization.
>
> > The tightness of the bound is not discussed by theory or experiments.
>
> Regarding the tightness of the theoretical bounds, we have added a new section (Appendix B.4) analyzing empirically the tightness of the bound on the total variation. We find that the observed increase in total variation late in training for some initializations is primarily driven by an increase in the Lipschitz constants of the layer-wise functions $g_l$, rather than a fundamental looseness in our bounds.
> We have empirically measured the Lipschitz constants during training, showing that the ratio between our theoretical upper bound and the observed values remains relatively stable during training.
> Previously we had compared the total variation late in training for some initialization scales, but observe that the theoretical result is a bound for the ratio of the Total Variation and a maximal Lipschitz constant of a subset of the layers. When this is considered, the theoretical bound tracks the experimental results as shown in Figure 9.
>
>
> We would also like to note that the bounds relating the local complexity and the rank of learned representations in Section 4 are controlled in the sense that we have both an upper and lower bound connecting the local complexity to the local rank. See Theorem 5, Equations (11) and (12).
>
> In the revision we have included further improvements, including Proposition 6, which directly relates the total variation and the existence of adversarial examples in a simplified setting. Please let us know if any aspects would benefit from further clarification.

---

> > ### Comment · Reviewer_iFkq · 2024-11-25
> >
> > I thank the authors for addressing my concerns. As for the tightness, the authors showed that the previous bound sometimes fails to explain total variation with local complexity. Appendix B.4 discusses this with the dynamical change of the Lipschitz constant term. By replacing the Lipschitz constant term with some operator norm of weight metrics, the bound becomes consistent with empirical results. I appreciate this analysis. The tightness issue was pointed out by several reviewers, including me, and I think the authors addressed it. I'm on the acceptance side.  However, because the next score has a jump of 6 -> 8 (no 7), I would retain my original score.

---

### Official Review · Reviewer_kjik · 2024-11-04

**Soundness:** 3
**Presentation:** 3
**Contribution:** 2
**Rating:** 6
**Confidence:** 3

**Summary:**

This work introduces a complexity measure of a neural network with ReLU activations, called local complexity.
Local complexity measures the density of linear regions in the neighbourhood of an input point.
Authors provide theoretical upper and lower bounds with respect to two other quantities.
First is local rank, also introduced by the authors, which connects to representation learning.
Second is total variation, which relates to robustness properties of the network.

Theoretical results are complemented by numerical experiments, where links between local complexity and local rank / total variation are studied.

**Strengths:**

Authors propose a clear mathematical quantity to compute local complexity of the network.
Furthermore, authors describe connections of local complexity of the network to representation learning and its robustness from theoretical viewpoint. Proofs are easy to follow, with some simple examples given to provide intuition.

Experimental results with visualization of the proposed complexity measure show that it captures the density of non-linearities. Authors are also clear with the possible limitations of the connection between metrics, by adding experimental results without clear correlations between, e.g., total variation and local complexity.

**Weaknesses:**

My main concern is that there is little discussion on the sharpness of the theoretical bounds. For example, lower bounds in Theorems 5,6 depend on $B$, number of active neurons, but estimates for $B$ are only given at the initialisation, not after training. To better understand the theoretical bounds, one could plot them empirically. The fact that bounds might be loose can also be seen at the experiments, where the connection between the quantities (eg local complexity and total variation) is not clear, as the authors point out in Appendix C.

Also, as authors mention themselves, proof of Theorem 2 essentially directly follows from (Hanin & Rolnick 2019a).

**Questions:**

Is it possible to plot / show estimates of the bounds of Theorems 4,5?

What is $R(\mathcal{N}\_{\theta})$ in Proposition 7? According to definition in line 398-399, one always trivially has $R(\mathcal{N}\_{\theta}) = 0$.

Discussion in lines 200-210 is not very clear. For example, why noise is only added to biases, and not to the input / weight matrices?

Line 386: "our theoretical result involves several interdependent components": could you clarify this sentence?

---

> ### Author Response · Authors · 2024-11-22
> **Part I**
>
> We sincerely thank the reviewer for their thoughtful and detailed comments which have helped us improve our paper. Below we address each point raised:
>
> > Also, as authors mention themselves, proof of Theorem 2 essentially directly follows from (Hanin \& Rolnick 2019a).
>
> Our work uses a theoretical framework that shares similarities with the work of Hanin \& Rolnick, particularly in the application of the co-area formula to study linear regions. We believe that it is important to give credit to Hanin \& Rolnick for their ingenious application of the co-area formula to study linear regions. Kindly observe however that the co-area formula is a classic and general result in multivariate calculus and therefore that it will arise in all kinds of calculations involving a change of variables and results such as the Kac-Rice formula (see Appendix A.2 for simple example calculations of the relating our use of the co-area formula and the change of variables).
>
> Although as we acknowledged the proof of Theorem 2 can be obtained using results from Hanin \& Rolnick, overall the two articles differ substantially in their focus and range of results. Hanin \& Rolnick primarily investigate how the depth and number of neurons affect the expected number of linear regions of a neural network for a generic distribution over the parameters.
> In contrast, our analysis explores how the  linear regions vary across different regions of the parameter space, that is, we attend to specific choices of parameters.
> Since other works have not focused on specific choices of parameters, they do not provide a framework to compare to. Furthermore, in our work and results we are interested in the properties of the linear regions that are local to particular regions of the input space, an aspect which is not discussed in the work of Hanin \& Rolnick.
>
> > Is it possible to plot / show estimates of the bounds of Theorems 4,5?
>
> Regarding the tightness of theoretical bounds, we have added a new section (Appendix B.4) analyzing empirically the tightness on the bound for the Total Variation.
> We find that the observed increase in total variation late in training for some initializations is primarily driven by an increase in the Lipschitz constants of the layer-wise functions $g_l$, rather than a fundamental looseness in our bounds.
> We have empirically measured the Lipschitz constants during training, showing that the ratio between our theoretical upper bound and the observed values remains relatively stable during training.
> Previously we had reported an increase in the total variation late in training for some initialization scales, but we should point out that the theoretical result is a bound for the ratio of the Total Variation and the maximal Lipschitz constant of a subset of the layers. When this is considered, the theoretical bound tracks the experimental results as shown in Figure 9.
>
> We would also like to note that the bounds relating the local complexity and the rank of learned representations in Section 4 are controlled in the sense that we have both an upper and lower bound connecting the local complexity to the local rank.
> See Theorem 5, Equations (11) and (12).
>
> > What is $R(\mathcal{N}_\theta)$  in Proposition 7?
>
> Concerning the definition of $R(\mathcal{N}\_\theta)$ , we have revised the definition to be more precise: $R(f) = \inf\_{\theta} {\|\theta\|\_F : \mathcal{N}\_\theta(x) = f(x) \text{ for all } x \in \Omega }$, where the infimum is taken over weights $\tilde{\theta}$ that preserve the function. This quantity is non-zero in general.

---

> ### Author Response · Authors · 2024-11-22
> **Part II**
>
> > Discussion in lines 200-210 is not very clear. For example, why noise is only added to biases, and not to the input / weight matrices?
>
> We have revised Section 3.1 to better explain our approach.
> Our theoretical framework is based on the analysis of the level at which a ReLU transitions from being inactive to being active.
> The biases directly control such transitions and that is why we focus on perturbations of the biases.
> It is possible to extend the definitions to include perturbations of the weights as well. Perturbations to the weight matrices introduce changes in the direction of the boundaries, which are naturally more complicated. We have added Figure 5 in the appendix to illustrate how adding perturbations to the weight matrices does not qualitatively change our measure of the local complexity.
>
> > Line 386: "our theoretical result involves several interdependent components": could you clarify this sentence?
>
> To clarify point 4 in your review, what we mean by "several interdependent components" is that several factors play a role in our analysis. We have reworded the discussion around line 386 to be more specific.
>
> In this context, we made additional improvements to convey the interaction between different quantities in our article. Concretely, we added Proposition 6, which directly relates the total variation and the existence of adversarial examples in a simplified setting.
> We also added Appendix A.10, which bounds the change in the local complexity when the change in the parameters is small. Please let us know if any aspects would benefit from further clarification.

---

> > ### Comment · Reviewer_kjik · 2024-11-24
> >
> > I thank the authors for the response and addressing my comments.
> > In the new Appendix B.4 the authors compare $LC$ with lower bound $A := \frac{TV L c^{\eta}_{\mathrm{bias}}}{\max \lVert W_1 \ldots W_L \rVert} $.
> > Two things are still not clear to me:
> > 1. what is the reason behind the choice $\eta = 1$? Since $LC$ does not depend on $\eta$ and $c^{\eta}_{\mathrm{bias}}$ decreases with $\eta$, one can take $\eta$ large enough to decrease $A$ below any value of $LC$. In particular, when $\sigma = 1$, $LC$ may be smaller than $A$ for $\eta$ small enough, based on values from Figure 9.
> > 2. what is the effect of $B$ after training in the lower bounds?

---

> > > ### Author Response · Authors · 2024-11-26
> > >
> > > Thank you for your ongoing conversation and review of our work. We address your concerns
> > >
> > > 1) The bound holds for all values of $\eta$, both in theory and in practice. We chose $\eta = 1$, because we found empirically that this maximizes the lower bound, which would make this as bound tight as possible. We can do this by solving for the maximum of an equation which looks roughly like the following (We will take $C =\mathbb E|\nabla z|$ and ignore some constants here for simplicity of notation),
> > > $$
> > > g(\eta) = e^{-\eta^2} (C - \frac{1}{\eta^2}).
> > > $$
> > > This has an optimum, $$\eta_* = \sqrt{\frac{1+\sqrt{1+4C}}{2C}}.$$ For the values of $C$ we observed empirically, the optimum $\eta_*$ was near $1$. In this way, the choice of constant $\eta$ represents the nearly the best possible bound. The choice of $\sigma = 1$ will effect both the constant terms in the bound, as well as how the local complexity is measured in practice.
> > >
> > > 2) We do not think that the term $B$ would have much effect in the lower bounds. For this term to be non-zero, there would have to exist a layer in the network such that every neuron is off when evaluated at $x$ for some $x$ in the data distribution. In our new empirical results in Appendix B.4 and Figure 10, we do not see that this occurs during training for networks of reasonable width, so $B$ is typically measured to be constant at $0$ during training. We have added a note to the main text to reflect this.

---

> > > > ### Comment · Reviewer_kjik · 2024-11-26
> > > >
> > > > I would like to thank the authors for addressing my concerns. I am raising my score and hope that the extended discussion on the empirical tightness of the bounds (eg how $\eta = 1$ was picked) will be included in the revised version.

---

> > > > > ### Author Response · Authors · 2024-11-27
> > > > >
> > > > > Thank you for your thoughtful review which has helped us improve the paper. We have added a note on the choice of $\eta$ as desired.

---

### Meta-Review · Area_Chair_fWxN · 2024-12-05

**Metareview:**

The paper introduces a geometric quantity referred to as local complexity (Eq. (3)), aimed at explaining and relating various aspects in the theory of ReLU networks, including feature learning, representation cost, and adversarial robustness. Reviewers appreciated the conceptual approach, which seeks to provide an intuitive framework for understanding key phenomena in deep learning. However, several major issues were raised. While some concerns, such as the novelty of the proposed theoretical framework (seen as closely related to prior work, e.g., Hanin & Rolnick), the sharpness of the provided bounds, and the significance of certain concepts introduced in the text, were resolved to a reasonable extent, key issues persisted despite considerable discussions with the authors. Specifically, concerns regarding the theoretical and empirical contributions not sufficiently supporting key claims in the paper were not fully addressed, leading reviewers to maintain a lukewarm overall assessment. The authors are encouraged to incorporate the important feedback given by the knowledgeable reviewers.

**Additional Comments On Reviewer Discussion:**

See above, as well as strong objections by some of the reviewers.

---

### Decision · Program_Chairs · 2025-01-22

Reject